# TAF1 plays a critical role in AML1-ETO driven leukemogenesis

Ye Xu [1,2], Na Man [1,3], Daniel Karl[1,3], Concepcion Martinez[1,3], Fan Liu [1,3], Jun Sun [1,3], Camilo Jose Martinez [1,3], Gloria Mas Martin [1,3], Felipe Beckedorff [1,4], Fan Lai[1,4], Jingyin Yue [1,4], Alejandro Roisman[1,4], Sarah Greenblatt[1,3], Stephanie Duffort[2], Lan Wang[1,3,5], Xiaojian Sun [1,3,6], Maria Figueroa[1,4], Ramin Shiekhattar[1,4] & Stephen Nimer[1,2,3]*

AML1-ETO (AE) is a fusion transcription factor, generated by the t(8;21) translocation, that functions as a leukemia promoting oncogene. Here, we demonstrate that TATA-Box Binding Protein Associated Factor 1 (TAF1) associates with K43 acetylated AE and this association plays a pivotal role in the proliferation of AE-expressing acute myeloid leukemia (AML) cells. ChIP-sequencing indicates significant overlap of the TAF1 and AE binding sites. Knockdown of TAF1 alters the association of AE with chromatin, affecting of the expression of genes that are activated or repressed by AE. Furthermore, TAF1 is required for leukemic cell self-renewal and its reduction promotes the differentiation and apoptosis of AE+ AML cells, thereby impairing AE driven leukemogenesis. Together, our findings reveal a role of TAF1 in leukemogenesis and identify TAF1 as a potential therapeutic target for AE-expressing leukemia.

[1] Sylvester Comprehensive Cancer Center, University of Miami, 1120 NW 14th St, Miami, FL 33136, USA. [2] Department of Medicine, Miller School of Medicine, University of Miami, 1120 NW 14th St, Miami, FL 33136, USA. [3] Department of Biochemistry and Molecular Biology, Miller School of Medicine, University of Miami, 1501 NW 10th Ave, Miami, FL 33136, USA. [4] Department of Human Genetics, Miller School of Medicine, University of Miami, 1120 NW 14th St, Miami, FL 33136, USA. [5] Institute of Health Sciences, Shanghai Institutes for Biological Sciences, Chinese Academy of Sciences, Shanghai Jiao Tong University School of Medicine, Shanghai, China. [6] State Key Laboratory of Medical Genomics, Shanghai Institute of Hematology, Rui Jin Hospital, Shanghai Jiao Tong University School of Medicine, Shanghai, China. *email: snimer@med.miami.edu

The t(8;21)(q22;q22) is the most commonly observed chromosomal translocation in acute myeloid leukemia (AML) patients; it generates the AML1-ETO (AE) fusion protein[1–4], which contains the N-terminal 177 amino acids of AML1 [also known as RUNX1 (runt-related transcription factor 1)] fused to nearly the entire ETO protein[1–4]. AE impairs myeloid differentiation and promotes the self-renewal of hematopoietic stem cells (HSCs)[5–7]; both are critical for AE-driven leukemia development. The importance of AE in leukemia development makes it an attractive therapeutic target[8–11], yet targeting it directly has been difficult[12].

Transcription is a highly regulated multiple-step process in eukaryotes starting with the assembly of a preinitiation complex (PIC). For RNA polymerase II-dependent transcription, PIC assembly involves the loading of activators at enhancers, the binding of TBP to TATA-containing promoters and the subsequent recruitment of TAF1 (also TAFII250), the largest subunit of the transcription factor IID (TFIID) complex. TAF1 serves as a bridge to bring 12 more TAF proteins to promoter regions[13]. Recently, the various modes of assembly of the PIC and combinations of TFIID components have been described as promoter-specific, tissue-specific or cell type-specific[14–17]. For instance, TAF1 is absent from human embryonic stem cells and its overexpression in those cells triggers their differentiation[14].

TAF1 also plays a critical role in the expression of genes involved in cell cycle and apoptosis[18,19]. TAF1 mutations have been found in solid tumors[20,21] and TAF1 overexpression is associated with both the progression of prostate cancer, and with castration resistance[22]. We previously showed that acetylation of lysine-43 (K43) on AE by p300 plays a critical role in AE-induced leukemia and that TAF1 preferentially interacts with acetylated AE peptides[23]. In the present study, we confirm that TAF1 is associated with AE in leukemia cells and that knockdown (KD) of TAF1 impairs the self-renewal and promotes the myeloid differentiation and apoptosis of AML cells, thereby impeding leukemia cell growth. Further, loss of TAF1 reduces the association of AE with chromatin and its ability to regulate the expression of both AE activated and repressed genes. Together, these results reveal a unique role of TAF1 in AE-driven leukemogenesis, which is distinct from its function in the PIC. Most importantly, the KD of TAF1 has little effect on normal HSCs, implying that TAF1 has the potential to serve as a therapeutic target for AE-expressing AML.

## Results

### TAF1 is required for the proliferation of AE-expressing cells.

TAF1 is overexpressed in many AML cell lines compared to its expression in CD34+ hematopoietic stem progenitor cells (HSPCs) isolated from human umbilical cord blood (CB) (Supplementary Fig. 1a). To elucidate the role of TAF1 in the proliferation of AE-expressing cells, we knocked down TAF1 in two human t(8;21) AML cell lines, Kasumi-1 and SKNO-1, using two different TAF1 shRNAs that reduce TAF1 expression, compared to a scrambled shRNA control (Fig. 1a, b, middle and right panels). KD of TAF1 by either shRNA blocks the proliferation of these cells (Fig. 1a, b, left panels), suggesting that TAF1 is critical for their growth. As a component of the TFIID complex, it is possible that TAF1 might be generally required for the cells to proliferate. To evaluate this possibility, we knocked down TAF1 in the non-AE-expressing K562 leukemia cell line and in human CD34+ CB cells. The reduction in TAF1 expression in these two cell types was comparable to that achieved in Kasumi-1 cells (Fig. 1c, middle and right panels and Fig. 1d, right panel); however, decreasing TAF1 expression had little effect on the proliferation of these cells (Fig. 1c, d, left panels). We also examined

the effect of TAF1 KD on the growth of OCI-AML3 cells and found a modest effect on their growth in liquid culture (Supplementary Fig. 1b), consistent with a recent CRISPR dropout screen report[24]. Thus, TAF1 appears to play a particular role in the growth of AE-expressing cells.

To assess the effect of TAF1 on cell proliferation, we labeled cells with BrdU and measured the cell cycle profile using flow cytometry. KD of TAF1 reduced the percentage of Kasumi-1 cells in the S phase, but had no effect on the percentage of K562 cells or CD34+ cells in the S phase (Fig. 1e–h). Thus, it appears that TAF1 is particularly critical for the proliferation of AE-expressing cells.

To determine whether KD of TAF1 triggers apoptosis in AE-expressing cells, Kasumi-1 cells and K562 cells were infected with scrambled shRNA or two TAF1-directed shRNAs and stained with Annexin V and 7-AAD. TAF1 KD increased the percentage of apoptotic Kasumi-1 cells but did not increase the apoptosis of K562 cells (Supplementary Fig. 1c–e). Thus, TAF1 depletion primarily impairs cell cycle progression and induces apoptosis in AE-expressing cells.

### TAF1 KD promotes myeloid differentiation and impairs self-renewal.

AE blocks the myeloid differentiation of human CD34+ CB cells[23]. To define the role of TAF1 in differentiation, we knocked down TAF1 in AE-expressing CD34+ cells and in AE conditional knock-in mouse bone marrow cells (Supplementary Fig. 2a–c). We examined Mac-1 (CD11b) expression in the human cells (Fig. 2a) and Mac-1 and Gr-1 expression in the mouse cells (Fig. 2b) and found that in both settings, the block in myeloid differentiation triggered by AE was partially relieved by TAF1 KD. In contrast, TAF1 KD had little influence on myeloid differentiation or the expression of these markers in cells lacking AE. Taken together, it appears that TAF1 participates in the AE-mediated block of myeloid differentiation.

To investigate the role of TAF1 KD in AE-expressing HSPC self-renewal, bone marrow cells were isolated from AE conditional knock-in mice and infected with scrambled shRNA or two TAF1 shRNAs. After confirming AE expression and TAF1 depletion (Supplementary Fig. 2a), we performed serial replating colony formation assays (Fig. 2c) and cobblestone area forming (CAFC) assays (Fig. 2d). TAF1 KD significantly impaired the increased serial replating capacity and CAFC formation driven by AE. We also examined AE-expressing CD34+ human CB cells and confirmed that KD of TAF1 (Supplementary Fig. 2c) impaired cobblestone area formation induced by AE in human cells as it did in mouse HSPCs (Fig. 2d, e). Clearly, these results indicate that TAF1 promotes AE-induced HSPC self-renewal.

### TAF1 KD blocks the proliferation and self-renewal of AE9a+ cells.

While AE is insufficient to induce leukemia in mice by itself[25,26], expression of the alternatively spliced form of AE, AE exon 9a (AE9a) does induce leukemia in mice[27,28]. To determine the impact of TAF1 KD on AE9a-expressing leukemia cells, we developed an AE9a+ luciferase+ cell line, and used secondary spleen leukemia cells as shown in Fig. 3a (and described in Methods). To determine whether KD of TAF1 affects the growth of leukemia cells in vitro, the same numbers of AE9a+ secondary spleen leukemia cells transduced with scrambled or TAF1-directed shRNAs were grown and cell numbers were counted on days 3, 5, and 7. As shown in Fig. 3c, depletion of TAF1 (Fig. 3b) impairs the growth of AE9a+ leukemia cells recapitulating the effects observed on AE-expressing cells. To determine the impact of TAF1 depletion on the colony formation and self-renewal of these cells, colony formation assays and CAFC assays were performed using secondary spleen leukemia cells infected with

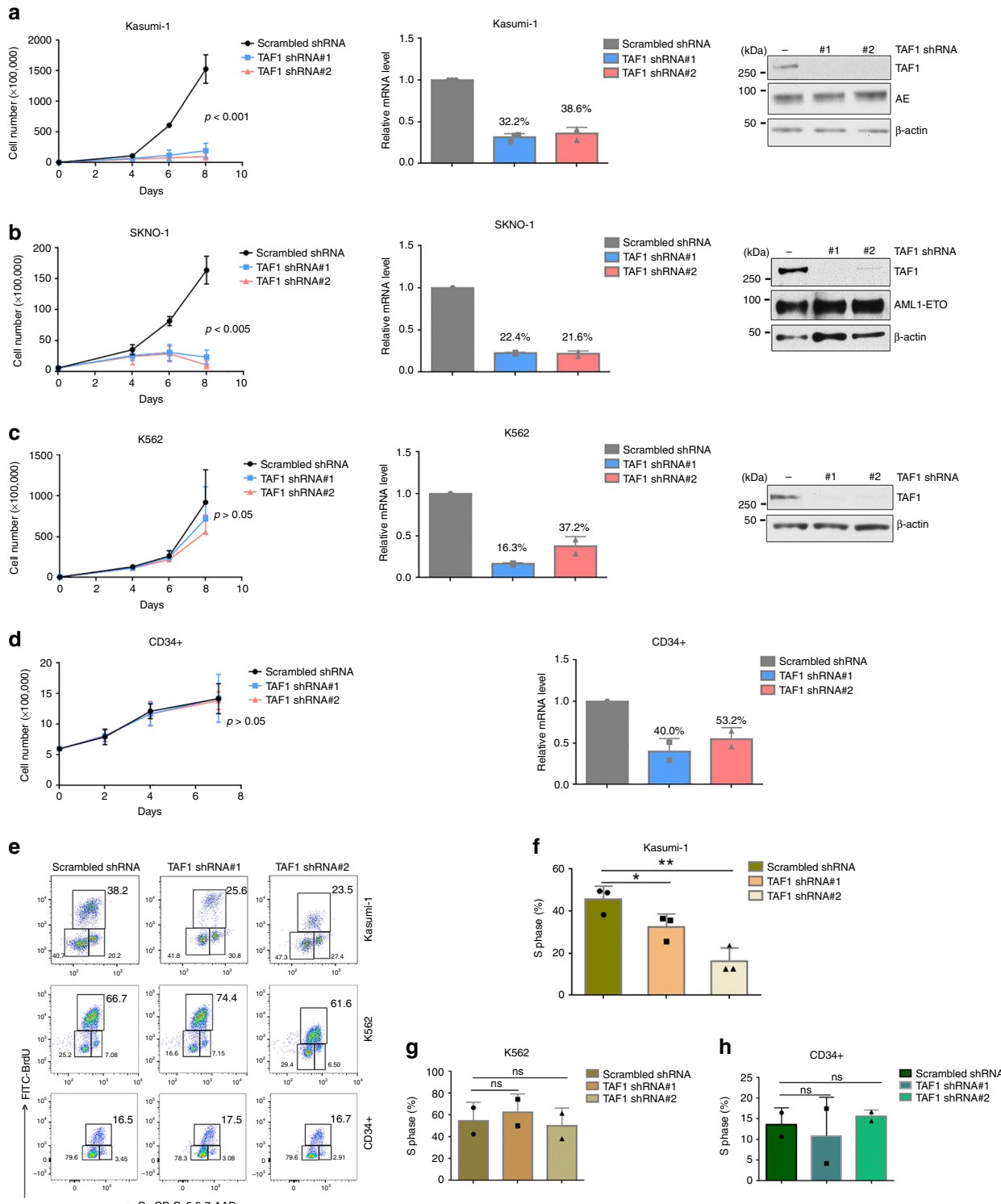

**Fig. 1** Depletion of TAF1 blocks the proliferation of AE-expressing cells. **a–d** Knockdown of TAF1 blocks the growth of Kasumi-1 cells (**a**) and SKNO-1 cells (**b**) and has little effect on the growth of K562 cells (**c**) or CD34+ cells (**d**). Kasumi-1 cells, SKNO-1 cells, K562 cells, and CD34+ cells were infected with scrambled shRNA or TAF1-directed shRNAs. The levels of TAF1 mRNA and TAF1 protein in each type of cells infected with scrambled shRNA or two different TAF1-directed shRNAs are shown in bar graphs and western blots. The TAF1 expression levels after knockdown are indicated as percentage above each column. The cell numbers between cells infected with scrambled shRNA and cells infected with TAF1 shRNAs at last time point were compared using Student $t$-test. $P$-values are displayed. **e–h** Knockdown of TAF1 reduces the percentage of Kasumi-1 cells in the S phase and has no influence on K562 and CD34+ cells. Cells were infected with scrambled shRNA or TAF1 shRNAs for 4 days and subjected for BrdU assay. Representative flow cytometry pictures are shown in **e**. **f–h** The percentages of Kasumi-1 cells (**f**), K562 cells (**g**), and CD34+ cells (**h**) with normal or reduced TAF1 levels in the S phase are shown in bar graphs. All experiments were repeated at least two times independently, **a** $n = 3$, b–d $n = 2$, **f** $n = 3$, **g–h** $n = 2$. All error bars represent the mean ± SD. The percentage of cells in the S phase in TAF1 shRNA-infected cells was compared with that in scrambled shRNA-infected cells. $P$ values were determined by Student's $t$-test. ns represents no significant difference, $*p < 0.05$, $**p < 0.01$

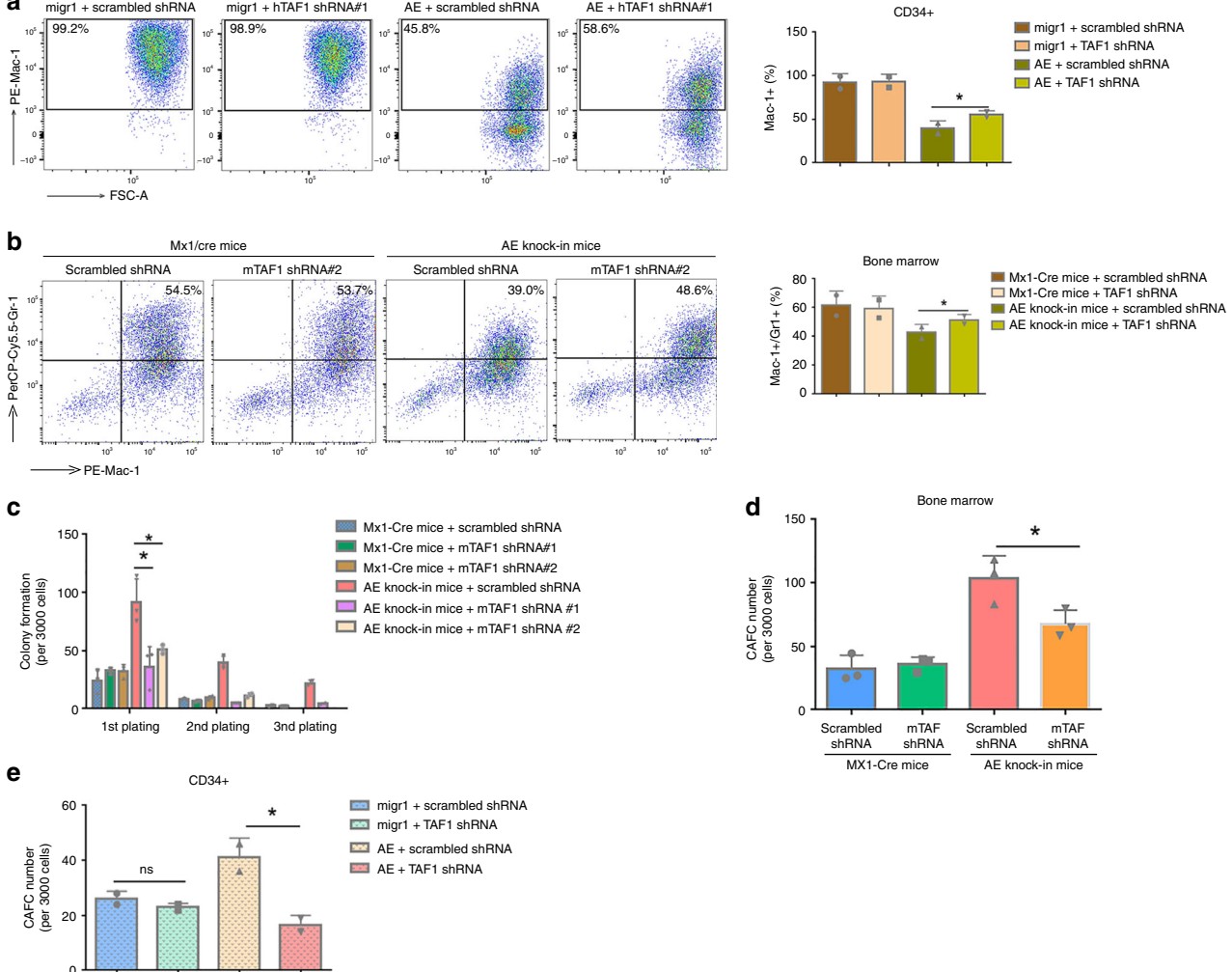

**Fig. 2** TAF1 depletion promotes myeloid differentiation and impairs self-renewal. **a, b** Knockdown of TAF1 partially reverses the blockade of myeloid differentiation driven by AE in human CD34+ cells (**a**) and mouse bone marrow cells (**b**). To monitor myeloid differentiation, CD34+ cells were incubated in myeloid differentiation promoting medium for 4 days. Mac-1 was used as the myeloid marker for the human CD34+ cells, while Mac-1 and Gr-1 were used for mouse bone marrow cells isolated from Mx1-Cre or AE knock-in mice. **a, b** Representative flow cytometry pictures are shown on the left. Each Bar graph is the summary of at least two independent experiments using either CD34+ cells or bone marrow cells. * represents the comparison of the relative expression levels of differentiation marker(s) between AE group with normal TAF1 level and AE group with reduced TAF1 levels by Student's t-test and p < 0.05. **c** Knockdown of TAF1 reduces the self-renewal induced by AE. Serial plating assays were performed using bone marrow cells isolated from Mx1-Cre or AE knock-in mice and transduced with scrambled shRNA or TAF1-directed shRNAs. The numbers of colonies at each plating are shown as mean ± SD. **d, e** Knockdown of TAF1 impairs cobblestone area forming cell (CAFC) frequency in AE+ bone marrow cells (**d**) and CD34+ cells (**e**). The numbers of cobblestone area were counted at week 5 and shown as mean ± SD. The colony numbers in cells expressing AE with normal or reduced TAF1 levels were compared. *p < 0.05, ns indicates p > 0.05. P values were determined by Student's t-test. All experiments were repeated at least two times independently. **a, b** n = 2, **c, d** n = 3, **e** n = 2. All error bars (**a-e**) represent the mean ± SD

scrambled or TAF1-directed shRNAs. As shown in Figs. 3d, e, TAF1 KD reduces the colony formation of leukemia cells and the numbers of cobblestone area forming cells (CAFCs) indicating that TAF1 is critical for maintaining the self-renewal and frequency of AE9a+ leukemia stem cells. These data reveal that TAF1 is indispensable for both AE-expressing HSPCs and AE9a-expressing leukemia cells.

**TAF1 plays a pivotal role in AE9a-induced leukemogenesis**. To determine whether TAF1 is involved in AE9a-driven leukemogenesis in vivo, we knocked down TAF1 in the AE9a+ luciferase+ mouse bone marrow cell line using two distinct TAF1 directed shRNAs. After confirming TAF1 depletion (Supplementary Fig. 3b), we injected equal numbers of AE9a+ luciferase + cells with normal or KD levels of TAF1 into irradiated recipient

mice. TAF1 KD had no influence on the engraftment of the AE9a+ luciferase+ cells (Supplementary Fig. 3a). However, mice injected with TAF1 KD AE9a+ cells survived substantially longer than mice injected with AE9a+ cells that express wild-type levels of TAF1 (Fig. 4a), implying a pivotal role for TAF1 in leukemia development. Further, we examined the growth of AE9a+ leukemia cells in mice using the IVIS imaging system: 20 days after injection of the AE9a+ luciferase+ cells, the luciferase signal was widely distributed in the spleen and bone marrow of 6/8 recipient mice. In contrast, only 1/16 mice that received TAF1 KD AE9a+ luciferase+ cells had detectable luciferase signal at that time (Fig. 4b, c). To further demonstrate that TAF1 loss impairs the growth of leukemia cells in mice, we quantified the number of GFP tagged AE9a+ luciferase+ cells in the peripheral blood 3 weeks after injection using flow cytometry and found far fewer

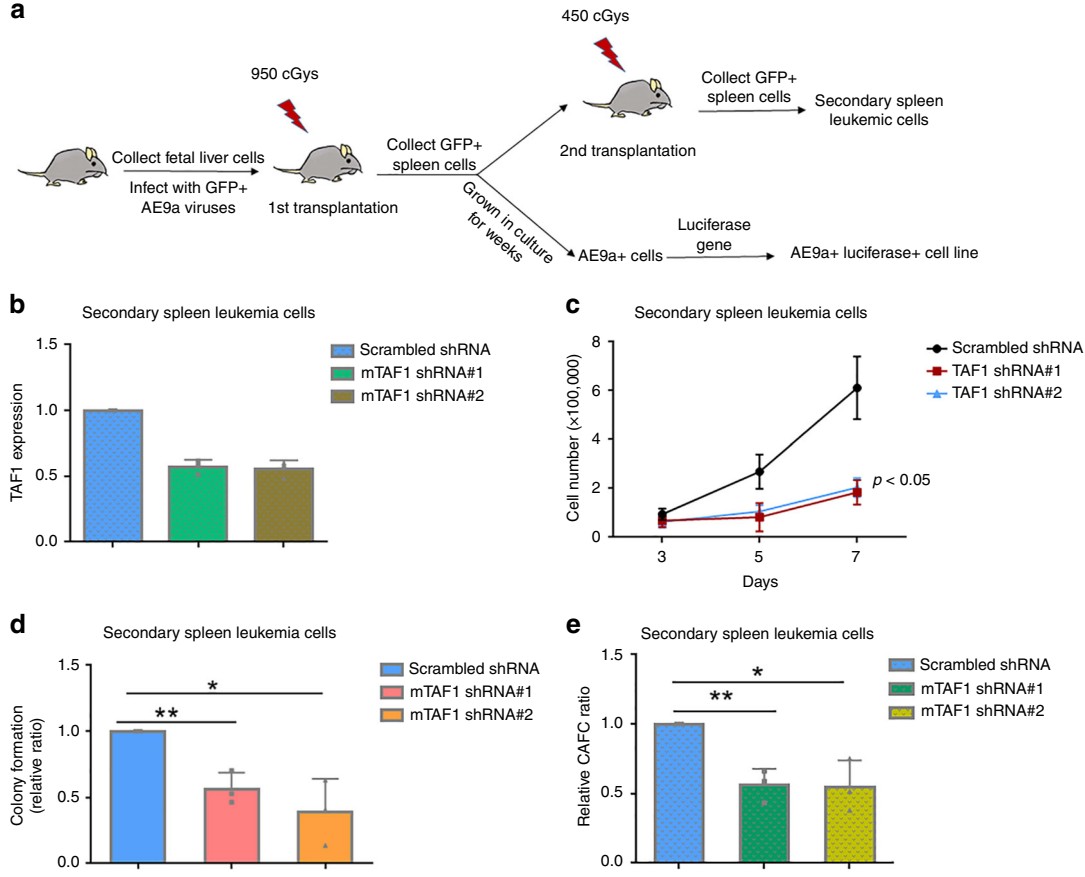

**Fig. 3** Depletion of TAF1 impairs the proliferation and self-renewal of AE9a+ cells. **a** Schema of the generation of AE9a+ luciferase+ cell line and the collection of secondary spleen leukemia cells. **b** TAF1 depletion levels in secondary spleen leukemia cells. Secondary spleen leukemia cells were infected with scrambled shRNA or TAF1 direct shRNAs for 3 days and mRNA was extracted. **c** Knockdown of TAF1 significantly reduces the proliferation of secondary spleen leukemia cells. All groups started with the same cell numbers; cell numbers were counted on days 3, 5, and 7. The cell numbers between cells infected with scrambled shRNA and cells infected with TAF1 shRNAs at day 7 were compared using Student's $t$-test. **d** TAF1 is critical for the colony formation of secondary spleen leukemia cells. The colony number in each TAF1 KD group was compared with that in scrambled group and the relative ratios are displayed. **e** TAF1 is important for maintaining CAFC frequency in secondary spleen leukemia cells. The number of cobblestone areas was counted at week 5. The CAFC number in each TAF1 KD group was compared with that in scrambled group and the relative ratios are displayed. *$p < 0.05$, **$p < 0.01$, $p$ values were determined by Student's $t$-test. **b–e** Experiments were repeated three times independently. All bars represent the mean ± SD

TAF1 KD GFP+ cells (Fig. 4d). We also infected AE9a+ leukemia cells obtained from the secondary transplantation of splenic leukemia cells with either scrambled or TAF1-directed shRNAs before injecting them into recipient mice. After tertiary transplantation, the mice receiving TAF1 KD cells (Supplementary Fig. 3c) had longer survival (Fig. 4e) with far fewer GFP+ AE9a+ cells in the peripheral blood (Fig. 4f, Supplementary Fig. 3d). Together, these data indicate that TAF1 contributes critically to AE9a-induced leukemogenesis, and could serve as a potential target for anti-leukemia therapy.

**TAF1 is associated with AE**. We initially identified TAF1 based on its binding to an acetylated AE peptide in a peptide pull down assay[23]. To determine whether the endogenous TAF1 protein associates with full-length AE in leukemia cells, we performed reciprocal co-immunoprecipitations using anti- TAF1 and anti-ETO antibodies. As shown in Fig. 5a, b, we readily detected the physical interaction of TAF1 with AE in Kasumi-1 cells, while no association was seen in K562 cells, as expected (Fig. 5a). We performed mass spectrometry following TAF1 immunoprecipitation using Kasumi-1 cells, and confirmed that TAF1 not only binds to AE, CBFβ and p300, but also binds to AE cofactors such

as SIN3A and HDAC1 (Table 1). To examine whether other components of TFIID are present in the TAF1/AE complex, we performed co-immunoprecipitation assays using an anti-ETO antibody. As shown in Supplementary Fig. 2d, TAF15 was detected in the TAF1/AE complex. However, mass spectrometry analysis indicates that except for TAF15 other components of the TFIID complex are not detected (Supplementary Table 1). Since acetylation of lysine-43 (K43) on AE is critical for AE-driven leukemogenesis[23], we examined whether acetylation of K43 is needed for TAF1 binding to AE. We co-transfected full length AE, AE K43R, AE K24R or AE K24RK43R with TAF1 and p300 into 293 T cells and performed co-IPs using an anti-TAF1 antibody. As shown in Fig. 5c, mutating lysine-43 in AE to arginine abrogates the interaction of AE with TAF1 (lanes 3 and 5), while mutating lysine 24 to arginine had a modest influence on their association; thus, lysine-43 acetylation appears to be critically important for the interaction of AE with TAF1. To determine whether the bromodomains in TAF1 are critical for recognizing K43 acetylation on AE, we deleted both bromodomains of TAF1 and found that this abrogated the association of TAF1 with AE (Fig. 5d). Thus, it appears that TAF1 binds to acetylated lysine-43 on AE through its bromodomains.

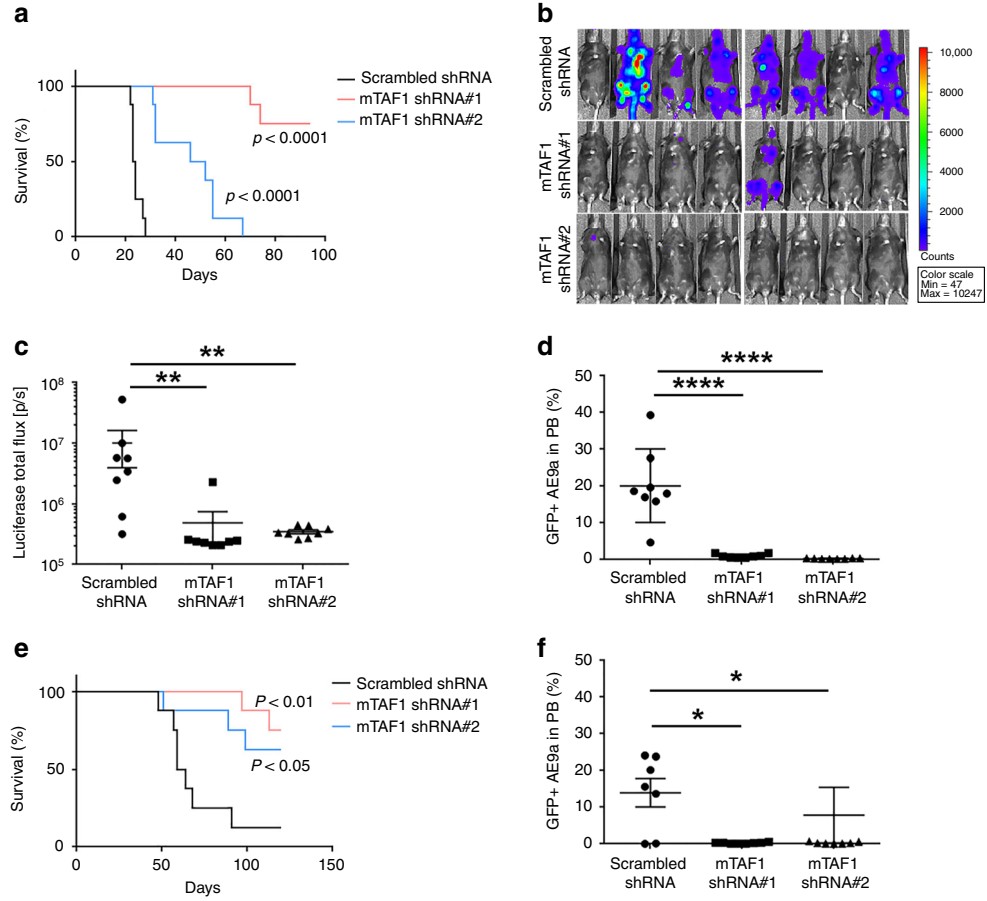

**Fig. 4** TAF1 knockdown significantly delays leukemia development. **a** Kaplan–Meier plots show that TAF1 KD significantly extends the survival of recipient mice transplanted with AE9a+ luciferase+ cells. Sublethally irradiated C57BL/6J mice were injected with AE9a+ luciferase+ cells transduced with scrambled shRNA or mouse TAF1 shRNAs ($n = 8$ in each group). p value was determined using Log-rank (Mantel–Cox) test. **b** In vivo luciferase imaging indicates that knockdown of TAF1 remarkably impairs leukemia development ($n = 8$ in each group). Mice were injected with AE9a+ luciferase+ cells expressing wild-type level or reduced levels of TAF1. Twenty days after transplantation, IVIS imaging was performed. **c** The quantification of total luciferase signal in each mouse of each group as shown in (**b**). The luciferase signal in the TAF1 KD group was compared to the signal in the scrambled shRNA control group. **d** KD of TAF1 reduces the presence of GFP+ AE9a+ luciferase+ cells in the peripheral blood. Mice were injected with GFP+ AE9a+ luciferase+ cells infected with scrambled shRNA or TAF1 shRNAs. The percentage of GFP+ AE9a+ luciferase+ cells in peripheral blood of each mouse was measured 3 weeks after transplantation. The percentage of GFP+ AE9a+ in the peripheral blood of the TAF1 KD group was compared with the percentage of GFP+ cells in the scrambled shRNA group. P values were determined by Student's t-test. **e** Survival curves of mice injected with secondary spleen leukemia cells transduced with scrambled shRNA or TAF1 directed shRNAs. $n = 8$ mice in each group. P value was determined using Log-rank (Mantel–Cox) test. **f** The percentage of GFP+ AE9a+ cells in the peripheral blood of each mouse after receiving secondary spleen leukemia cells infected with scrambled shRNA or TAF1-directed shRNAs. Peripheral blood was collected 48 days after transplantation. The percentage of GFP+ AE9a+ cells in peripheral blood in the TAF1 KD group was compared with the percentage for the scrambled shRNA group. P values were determined by Student's t-test. *$p < 0.05$, **$p < 0.01$, ****$p < 0.0001$

**The expression of AE target genes is affected by depletion of TAF1**. Given the importance of TAF1 in mediating the effects of AE in AML cells, we explored how KD of TAF1 affects AE-regulated gene expression. *ID1* and *MYC* are AE activated genes, and we confirmed that their expression was reduced by AE KD in Kasumi-1 cells (Fig. 6a). Next, we showed that TAF1 KD also significantly reduced the expression of these genes without reducing the level of AE expression (Fig. 6b, d). We also used the AE9a+ mouse cell line, and found that depletion of TAF1 impairs the expression of *ID1 and MYC* (Fig. 6c). To exclude the possibility that KD of TAF1 impacts RNA polymerase II-dependent transcription globally, we compared a panel of RNA Polymerase II-dependent housekeeping genes, such as *ACTB* and *GAPDH*, and cell cycle regulatory genes, such as *CCND2*, *CCND3*, and *CDKN2D*, in TAF1 KD conditions using the RNA Polymerase I-dependent ribosomal transcript 18S as internal control. We found that TAF1 KD does not broadly influence RNA Polymerase II-

dependent transcripts, suggesting that residual TAF1 in our system was sufficient for its role as a general transcription factor (Supplementary Fig. 4c). Thus, AE works in concert with TAF1 to activate its target genes. To identify those genes regulated by both TAF1 and AE across the genome, we performed RNA-seq in Kasumi-1 cells (Supplementary Fig. 4a, b) and found that 36% (1413/3979) of AE-activated genes are also downregulated by the KD of TAF1 (Fig. 6e). KEGG analysis indicates that these genes are involved in DNA replication, RNA splicing, RNA transport, cell cycle, nucleotide metabolism, and ribosome biogenesis. Surprisingly, TAF1 KD also upregulates many genes including 26% of the genes that are upregulated following AE KD (Fig. 6f). KEGG analysis revealed that the AE repressed genes are involved in lysosome, FOXO signaling pathways, and glycosaminoglycan degradation (Fig. 6f). Among the upregulated genes in these pathways is *GADD45* which acts to promote apoptosis[29], and *HEXB*, which is a lysosomal enzyme. The upregulation of

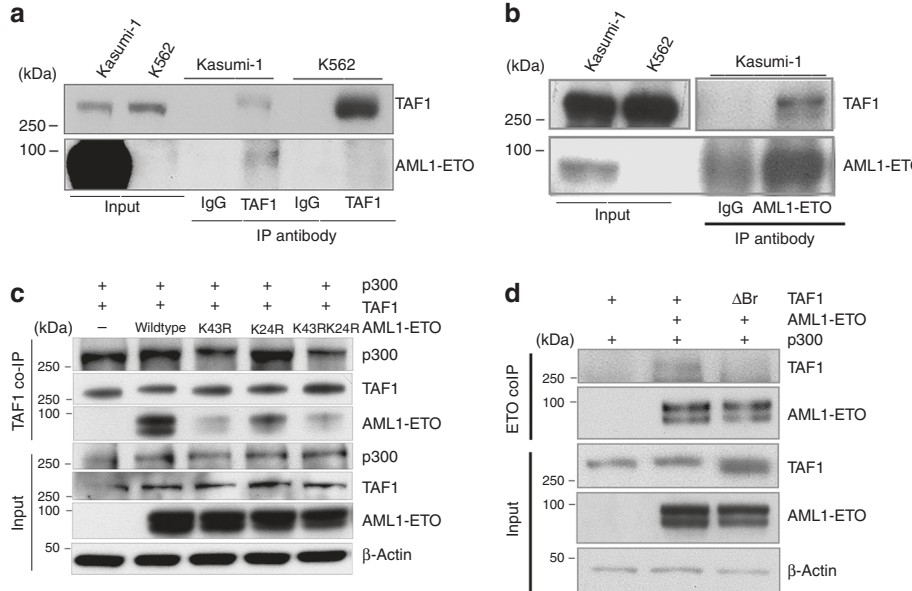

**Fig. 5** TAF1 associates with acetylated K43 on AE through its bromodomains. **a** TAF1 physically interacts with AE in Kasumi-1 cells. Co-immunoprecipitation was performed using anti-TAF1 antibody or normal mouse IgG. **b** Co-immunoprecipitation of TAF1 with AE using anti-ETO antibody or normal goat IgG. **c** Mutation of lysine-43 to arginine in AE reduces the interaction of AE with TAF1. 293T cells were transfected with p300, TAF1 and AE or AE mutants. Co-immunoprecipitation was performed using an anti-TAF1 antibody. **d** The deletion of the TAF1 bromodomain regions impairs its binding to AE. 293T cells were transfected with p300, AE, and TAF1 wild type or bromodomain deletion (ΔBr) plasmids. Co-immunoprecipitation was performed using an anti-ETO antibody

| Table 1 Mass spectrometry analysis of proteins associated with TAF1 | | | | |
|---|---|---|---|---|
| **Unique** | **Total** | **Reference** | **Gene symbol** | **AVG** |
| 6 | 20 | Q06455_MTG8_HUMAN | RUNX1T1 | 3.0288 |
| 4 | 8 | Q13951_PEBB_HUMAN | CBFB | 3.1504 |
| 6 | 7 | Q96ST3_SIN3A_HUMAN | SIN3A | 3.2562 |
| 5 | 12 | Q13547_HDAC1_HUMAN | HDAC1 | 3.0359 |
| 2 | 2 | Q09472_EP300_HUMAN | EP300 | 2.9337 |
| 23 | 46 | P26358_DNMT1_HUMAN | DNMT1 | 2.7142 |
| 5 | 10 | Q01196_RUNX1_HUMAN | RUNX1 | 3.1638 |

lysosomal enzymes may also be correlated with the apoptosis induced by TAF1 KD[30]. Clearly, TAF1 is essential for the expression of a subset of both AE activated and repressed genes implying the unique role of TAF1 in regulating AE target gene expression.

**TAF1 is critical for AE binding to chromatin**. To evaluate the role of TAF1 in the deposition of AE on chromatin, we generated subcellular fractions of Kasumi-1 cells with or without TAF1 KD. As shown in Fig. 7a and Supplementary Fig. 4d, the KD of TAF1 released AE from the chromatin fraction without reducing the overall level of AE in the cell. TAF1 KD does not change the localization of PU.1 with chromatin (Fig. 7a) implying the influence of TAF1 KD on AE binding at chromatin shows specificity. To investigate whether TAF1 and AE co-localize at AE target genes across the genome, we performed ChIP-seq using an anti-AML1-ETO antibody and an anti-TAF1 antibody in Kasumi-1 cells. As shown in Fig. 7b, AE peaks are widely spread at distal intergenic regions, promoter regions, and introns, while the majority of TAF1 peaks are at promoter regions, likely reflecting the function of TAF1 in PIC. When we examined the co-localization of TAF1 and AE peaks, we found that 58% of

TAF1 peaks overlap with AE peaks, while 19% of AE peaks overlap with TAF1 peaks (Fig. 7c). KEGG analysis indicates that the overlapping peaks are adjacent to genes linked to acute and chronic myeloid leukemia and pathways in cancer. The majority of overlapping AE and TAF1 peaks are located adjacent to transcription start sites (TSS) (Fig. 7d). However, a substantial number of overlapping peaks are seen ≥1 kb from TSS (non-TSS) (Fig. 7e). Approximately 60% of the TAF1/AE overlapping peaks at non-TSS are located at putative enhancers (Fig. 7e). AE/TAF1 co-bound peaks at non-TSS are adjacent to genes involved in pathways in cancer and AML (Supplementary Fig. 4e) while AE unique peaks at non-TSS regions are not directly involved in these pathways (Supplementary Fig. 4f) implying the importance of the TAF1/AE complex in these pathways. To confirm that TAF1 KD affects AE binding globally, we performed ChIP-seq using an anti-AE antibody in Kasumi-1 cells with normal levels or KD levels of TAF1 and found that TAF1 KD significantly reduces AE binding at those TAF1 and AE shared binding sites (Supplementary Fig. 4g). To exclude that TAF1 KD alters global chromatin accessibility, we performed ATAC-seq in Kasumi-1 cells with normal or reduced levels of TAF1 and found that TAF1 KD does not globally affect chromatin accessibility at TSS or enhancer regions (Supplementary Fig. 4h). Given that the acetylation of AE by p300 is critical for the interaction of TAF1 and AE, we examined whether p300 also shares occupancy with TAF1 and AE. As shown in Fig. 7f, 28% of AE/TAF1 co-occupant regions also have p300 bound such as the enhancer of the *ID1* gene (Fig. 7g). The combined analysis of ChIP-sequencing and RNA-sequencing data demonstrates that 36% of AE and TAF1 upregulated genes and 40% of AE and TAF1 repressed genes have overlapping TAF1 and AE peaks at their promoter and gene body (Supplementary Fig. 4i, j) indicating that these genes are likely to be directly controlled by both AE and TAF1. KEGG analysis indicates that these AE and TAF1 upregulated genes are related to cell cycle, spliceosome, and metabolism (Supplementary Fig. 4i), while the AE and TAF1 repressed genes, such as *HEXB* and

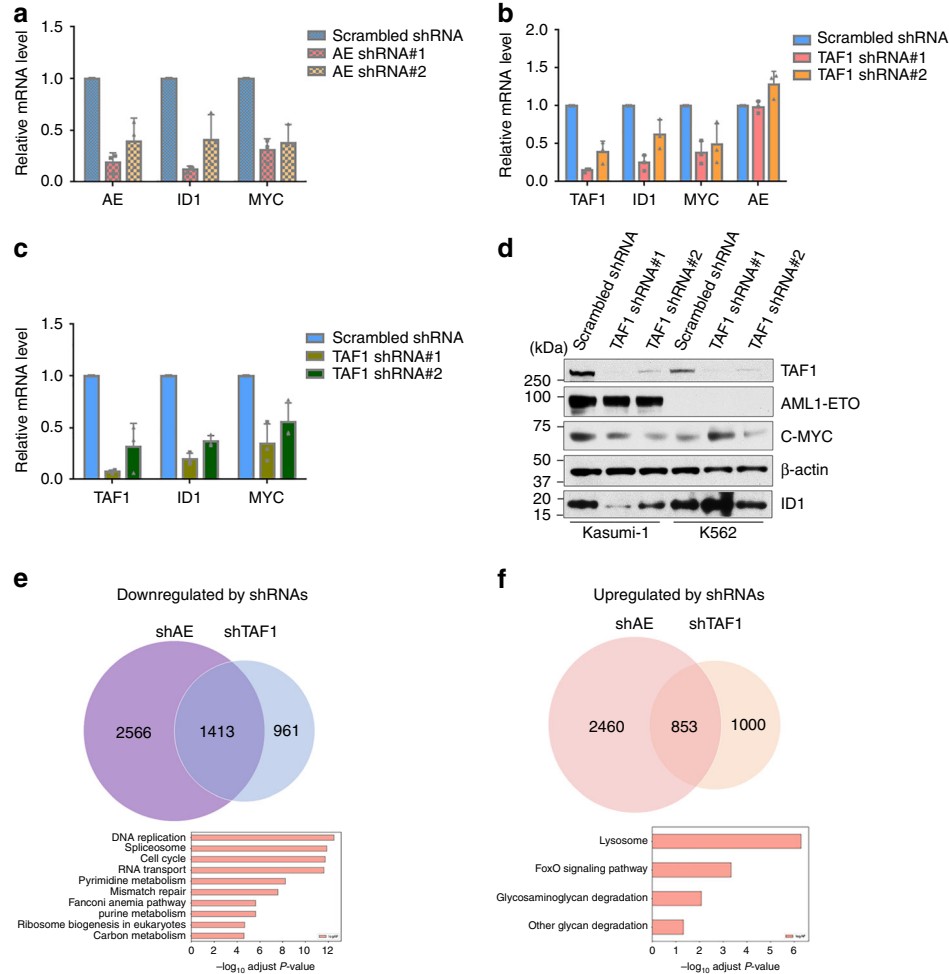

**Fig. 6** TAF1 knockdown affects the expression of AE target genes. **a**, **b** The influence of AE knockdown (**a**) or TAF1 knockdown (**b**) on the expression of AE target genes in Kasumi-1 cells. Kasumi-1 cells were transduced with scrambled shRNA or AE shRNAs (**a**) or TAF1 shRNAs (**b**) for 5 days. mRNA levels of individual genes were standardized by the level of 18S rRNA. **c** Depletion of TAF1 impairs the expression of AE target genes in AE9a+ cells. AE9a+ cells were transduced with scrambled shRNA or TAF1 shRNAs for 5 days and the mRNA levels of individual genes were measured and standardized by mouse 18S rRNA levels. **d** The protein levels of AE target genes in Kasumi-1 cells and K562 cells infected with scrambled shRNA or TAF1 shRNAs. **e**, **f** Venn diagrams of the numbers of genes downregulated (**e**) or upregulated (**f**) ($q < 0.05$) in Kasumi-1 cells after TAF1 KD (shTAF1) or AE KD (shAE). Scrambled shRNA-infected cells were used as controls. Kyoto Encyclopedia of Genes and Genomes (KEGG 2016) gene set analysis was performed on genes downregulated (**e**) or upregulated by both TAF1 KD and AE KD. **a–c** Experiments were repeated independently three times and all bars represent the mean ± SD

*GADD45*, are associated with the lysosome and FOXO signaling pathways (Supplementary Fig. 4j). Collectively, these data imply that the binding of TAF1 with AE at both promoter and enhancer regions is critical for the expression of a subset of AE target genes, which are implicated in cell cycle and leukemia.

**TAF1 inhibition reduces the proliferation of AE-expressing cells.** Bay-364 (also named Bay-299N) is a commercially available small-molecule inhibitor of the second bromodomain in TAF1, while Bay-299 is a bromodomain inhibitor of both BRD1 and TAF1. Given the essential role of TAF1 in the proliferation of AE-expressing cells, we examined the effects of Bay-364 and Bay-299 on the growth of Kasumi-1, K562 and CD34+ cells. As shown in Fig. 8a, d, Kasumi-1 cells are more sensitive to Bay-364 treatment than CD34+ cells or K562 cells, whose growth is minimally affected. In contrast, Bay-299 suppresses the growth of Kasumi-1 cells, but has a similar effect on CD34+ cells and K562 cells (Fig. 8b, d). We also used JQ-1, which inhibits the bromodomains of several BET family proteins including BRD2, BRD3, BRD4, and BRDT. JQ-1 is more potent than Bay-364 in suppressing the

growth of Kasumi-1 cells (Fig. 8c); however, it has similar effects on Kasumi-1 cells and CD34+ cells, likely indicative of its broader effect (Fig. 8c, d). Because Bay-364 selectively suppresses the proliferation of AE-expressing cells with much less toxicity on HSPCs, Bay-364 or similarly acting agents[31], might be a viable therapeutic option for AE-expressing leukemia.

Next, we examined the effect of Bay-364 on AE-mediated gene expression using Kasumi-1 cells treated with Bay-364 or vehicle for 3 days. Bay-364 decreased the expression of *ID1* and *MYC*, without affecting the expression of AE, similar to what was seen with TAF1 KD (Fig. 8e). To assess the importance of the second TAF1 bromodomain in controlling TAF1 and AE regulated genes, we compared the RNA-seq data from TAF1 KD and AE KD experiments with the RNA-seq data from Kasumi-1 cells, treated with or without Bay-364. In total, 1022 of 2642 (39%) AE and TAF1 regulated (both downregulated and upregulated) genes identified by TAF1 KD and AE KD are also controlled by the TAF1 bromodomain inhibitor (Fig. 8g, left panel). These genes play roles in cell cycle, DNA replication, metabolism, and apoptosis (Fig. 8g, right panel). In addition, Bay-364 treatment

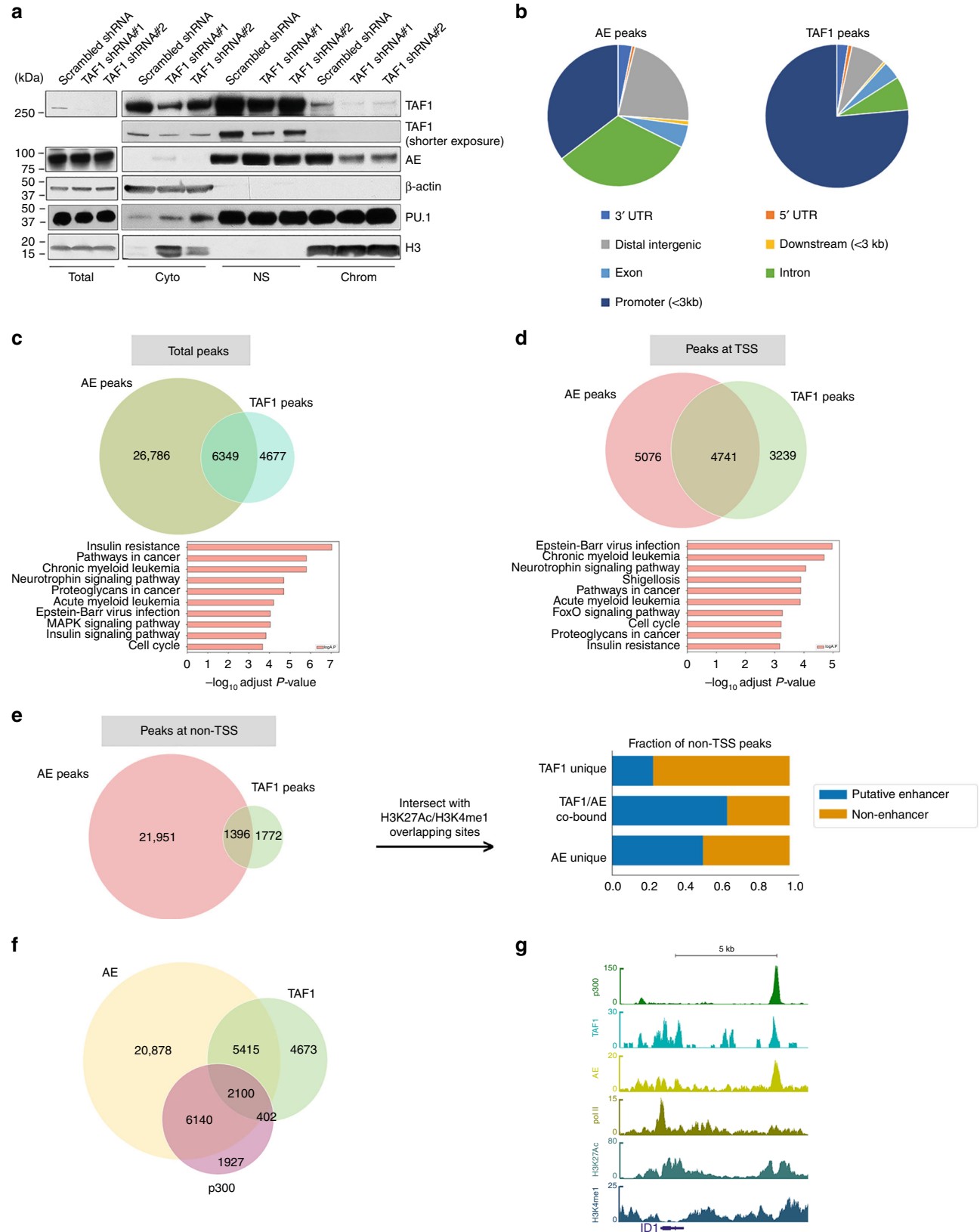

significantly impairs the colony formation of AE9a+ cells (Fig. 8f, Supplementary Fig. 3e). Taken together, the second TAF1 bromodomain appears to be selectively essential for the survival and colony formation of AE-expressing cells by regulating a critical subset of AE target genes.

## Discussion

TAF1 is a largest subunit in the PIC and its functions and enzymatic domains have been extensively studied in vitro[22,32–34]. These studies have identified a co-activator function for TAF1, in addition to its role in the PIC. In the present study, we were able

**Fig. 7** TAF1 is critical for the association of AE with chromatin. **a** Knockdown of TAF1 reduces the association of AE with chromatin in Kasumi-1 cells. Kasumi-1 cells were transduced with scrambled shRNA or TAF1 shRNAs for 5 days and then the subcellular fractionations were isolated. Total indicates the whole-cell lysate; cyto indicates the cytoplasm fraction; NS indicates the nuclear soluble fraction; chrom indicates the chromatin fraction. **b** Pie charts illustrate the distribution of AE peaks or TAF1 peaks across the genome. AE or TAF1 peaks were mapped to closest Refseq gene. Distal intergenic region is the region greater than 3 kb from either upstream or downstream of transcription start sites. The distance in the parentheses was measured from transcription start site. **c** Venn diagram illustrates the numbers of total AE peaks or TAF1 peaks or AE and TAF1 overlapping peaks in Kasumi-1 cells. Enrichment analysis for KEGG gene sets was performed on genes adjacent to overlapping TAF1 and AE peaks. **d**, **e** Venn diagram illustrates the numbers of AE peaks or TAF1 peaks or AE and TAF1 overlapping peaks at the TSS (within 1 kb of the transcription start sites) (**d**) ($p < 1.0e-5$) or at non-TSS regions (>1 kb of transcription start site including enhancers) (**e**) ($p < 1.0e-3$). $P$ values were estimated using a Monte Carlo simulation of shuffled peaks within either the TSS background or the non-TSS genomic background. The fractions of TAF1 unique peaks, TAF1/AE co-bound peaks, and AE unique peaks at putative enhancers or non-enhancers are plotted (**e**, right panel). Enhancers were defined as the regions with both H3K4me and H3K27Ac peaks excluding TSS regions. **f** Venn diagram illustrates the numbers of AE peaks, TAF1 peaks, p300 peaks, and their overlapping peaks. **g** The representative picture of the peaks of p300, TAF1, AE, polymerase II (pol II), histone H3 lysine 27 acetylation (H3K27Ac), and H3 lysine 4 monomethylation (H3K4me1) at AE-activated gene *ID1*

to define the genomic localization of TAF1 and AE, and the influence of TAF1 KD or TAF1 bromodomain inhibition on AE target gene expression using ChIP-seq and RNA-seq analyses. TAF1 co-localizes with AE at transcription start sites, and other regions of the genome, and TAF1 KD reduces AE binding to chromatin and impedes the expression of AE target genes such as *ID1* and *MYC*, indicating a role of TAF1 in controlling AE target gene expression. As predicted, we demonstrated that TAF1 plays a critical role in controlling AE upregulated genes. However, we also found 343 genes that are repressed by AE and TAF1, and that bind to AE and TAF1 in ChIP-seq studies, implying that TAF1 may also function as a corepressor for AE in negatively regulating gene expression. Although KD of TAF1 alters the expression of both AE upregulated and downregulated genes, it does not impact the general level of transcription (Supplementary Fig. 4c) implying that TAF1 functions as a specific co-activator and corepressor for AE target genes. Based on these findings, we propose a model for AE target gene expression (Fig. 8h), where the binding of TAF1 bromodomains to p300 acetylated AE directs the deposition of AE at a subset of AE target genes and allows AE to control its target gene expression. This role of TAF1, in regulating gene expression, is distinct from its role in pre-initiation complex formation.

We found a majority of TAF1 peaks at promoter regions, while a substantial fraction of AE peaks are located at either distal intergenic regions or introns (Fig. 7b). The shared binding sites of TAF1 and AE are found at both TSS ($p < 1.0e-5$) and non-TSS regions ($p < 1.0e-3$) implying that the association of TAF1 with AE is not restricted to promoter regions. In addition, TAF1 and AE shared binding sites at non-TSS are close to the genes involved in cancer or AML pathways, while the AE unique peaks (which are not overlapping with TAF1) at non-TSS are not directly related to these pathways (Supplementary Fig. 4e, f).

Co-immunoprecipitation and mass spectrometry studies identified proteins associated with AE and among the proteins contained in the TFIID complex, only TAF1 and TAF15 were found (Supplementary Fig. 2d, Supplementary Table 1). TAF15 is not a canonical component of TFIID and in fact, fusion of the TAF15 gene to the ZNF384 gene has been identified in patients with AML and acute lymphoblastic leukemia[35]. It is well known that TAF7 forms a subcomplex with TAF1 in the TFIID complex[36], yet TAF7 cannot be detected by either western blot or mass spectrometry in a complex with TAF1 in AE-expressing cells. In fact, studies using human embryonic stem cells have shown that the combination of TAFs in the PIC is both cell context and promoter dependent[14]. Although we cannot exclude that limitations in the affinity and specificity of the antibodies used or the co-immunoprecipitation protocol hampered our ability to identify other TAFs in the AE/TAF1 complex, it appears that few TAFs are engaged in the AE/TAF1 complex.

In our study, we have found that TAF1 is important for the binding of AE to its target genes (Fig. 7a, Supplementary Fig. 4d, g). However, the detailed mechanism behind how this occurs remains elusive. ATAC-seq data indicate that TAF1 KD does not alter global chromatin accessibility (Supplementary Fig. 4h), which would suggest that TAF1 promotes the more local functions of AE. CBFβ forms a heterodimer with of RUNX1, which enhances the binding of these proteins to their DNA consensus binding sequences[37]. The possibility that TAF1 influences the productive association of AE with CBFβ requires further investigation. As a scaffold protein, TAF1 is well known to mediate the docking of other TAFs within TFIID to promoter regions, through its N-terminal TBP binding sequence. It is possible that TAF1 helps provide a platform for other proteins, which could assist in the recruitment of AE to the chromatin. The crystal structure of TAF1 reveals that the winged helix domain in N-terminal of TAF1 could contribute to its DNA binding[38]. Furthermore, a second DNA-binding module at the C-terminal of TAF1 was identified[39]. In addition, TAF1 has multiple enzymatic domains. Given that AE has its own DNA-binding domain, in vitro assays to test whether theses domains (other than the bromodomains) participate in shaping the DNA binding activity or other activities of AE have not been revealing. Additional experimentation is needed to determine whether the TAF1 enzymatic domains or its two DNA-binding modules affect the in vivo association of AE with chromatin.

Targeting TAF1 impairs the proliferation of AE-expressing leukemia cells and their self-renewal, thereby delaying leukemogenesis. Although the effect of TAF1 KD in suppressing leukemic cell growth could be a general effect on cell growth, we found that TAF1 KD had little or no influence on the growth of K562 cells or CD34+ CB cells. Similarly, AE-expressing cells were substantially more sensitive to TAF1 bromodomain inhibition than K562 cells or human CD34+ CB cells. TAFs have recently emerged as potential therapeutic targets in leukemia; based on a CRISPR/Cas9 screen[24], TAF1 appears to be essential for MV4-11 AML cells (that express MLL-AF4), but dispensable for the growth of the MOLM-13, HL-60, OCI-AML3 AML cell lines and two different solid tumor cell lines[24]. Collectively, these data further highlight the cell-type-specific dependency on TAF1. In addition, TAF1 has been found in CBFβ-MYH11 and RUNX1 containing complexes[24,40], which may reflect an essential cooperativity of TAF1 with the activities of these oncogenic transcription factors or with the more fundamental mechanisms that they control.

Recently, the Vakoc lab described that TAF12, another component of the TFIID complex, acts as co-activator to protect the leukemogenic oncogene MYB, from undergoing protein degradation[41] in AML. Another study, using CRISPR dropout screening, identified requirements for individual members of the TFIID complex in the growth of some but not other human acute

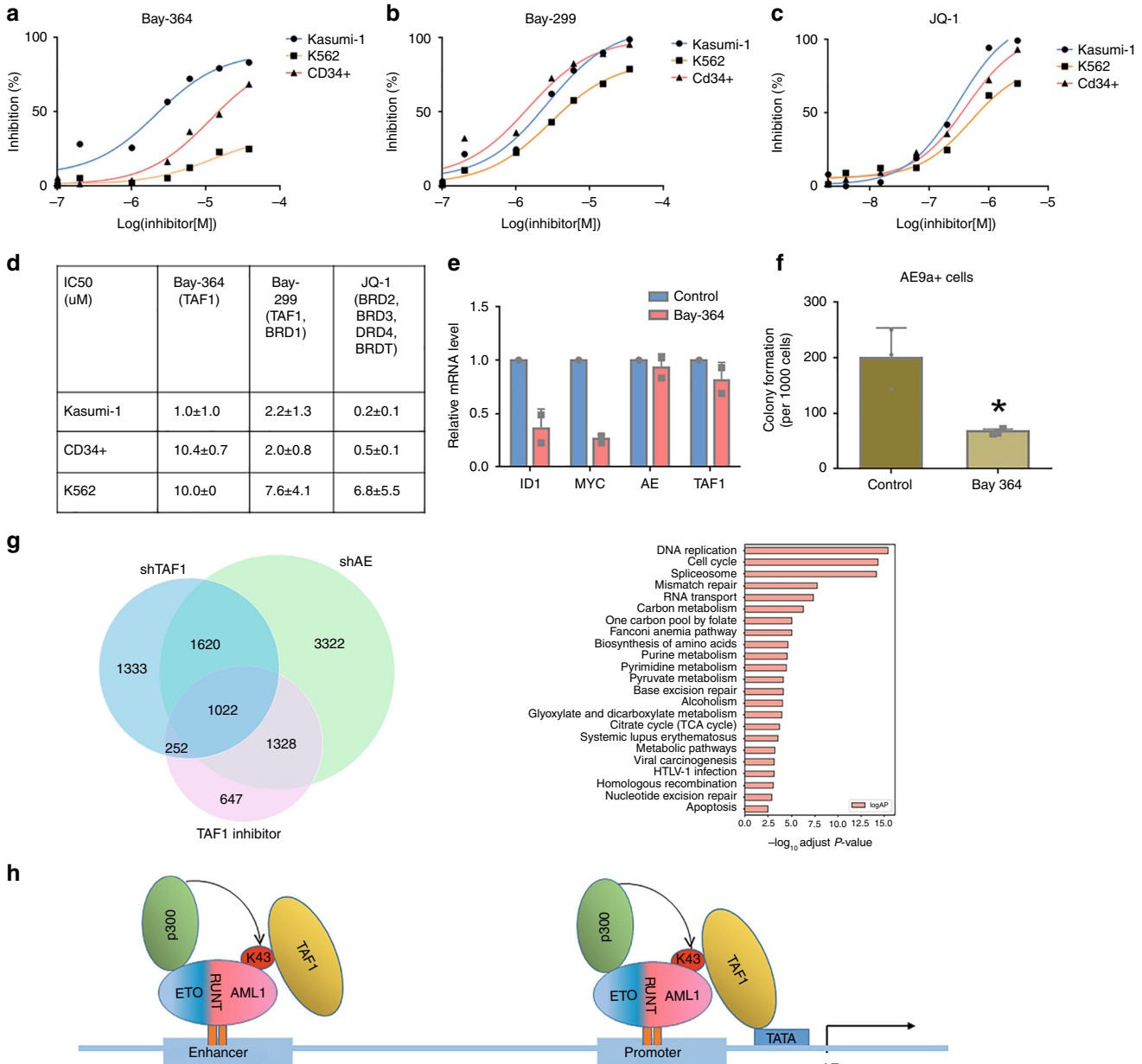

**Fig. 8** TAF1 bromodomain inhibition reduces the growth of AE-expressing cells. **a–c** The growth of Kasumi-1 cells, CD34+ cells, and K562 cells in the presence or absence of different concentrations of bromodomain inhibitors Bay-364, Bay-299, or JQ-1. The cell growth was measured by CellTiter-Glow luminescent cell viability assay after 3 days of treatment with the inhibitors or vehicle. Experiments were repeated three times and representative figures are shown. **d** IC$_{50}$s (μM) of each inhibitor in Kasumi-1, CD34+, and K562 cells. IC$_{50}$s were calculated from three independent experiments using GraphPad Prism. Targeted proteins by individual inhibitor are shown in parentheses. **e** TAF1 inhibitor Bay-364 represses the expression of AE upregulated genes. RNA was extracted from Kasumi-1 cells treated with vehicle or Bay-364 for 72 h. mRNA levels of individual genes were standardized by 18S rRNA level. **f** TAF1 inhibitor Bay-364 abrogates colony formation in AE9a+ cells. AE9a+ cells were treated with vehicle or 10 μM Bay-364 for 2 days and then 1000 AE9a+ cells were plated on methylcellulose-based medium containing vehicle or Bay-364. Colonies were counted 7 days after plating. The colony numbers from Bay-364-treated cells were compared to the colony numbers from the vehicle-treated cells and p value was determined by Student's t-test. *p < 0.05. **g** Venn diagrams illustrate the overlapping genes differentially expressed (q < 0.05) after TAF1 KD (shTAF1), AE KD (shAE), or TAF1 inhibitor Bay-364 treatment in Kasumi-1 cells. KEGG analysis was performed on 1022 genes differentially expressed after TAF1 KD, AE KD, and Bay-364 treatment. **h** TAF1 working model on AE target genes. p300 acetylates AE lysine-43 and the acetylated AE can be recognized by bromodomain of TAF1. TAF1 facilitates the association of AE to either the promoter or enhancer of a subset of AE target genes and then transcription is either activated or repressed. **e**, **f** Experiments were repeated independently at least two times. **e** n = 2, **f** n = 3. All bars represent the mean ± SD

leukemia cells[24]. Together with our study of TAF1 in AE-driven AML, these findings demonstrate that TAFs not only participate in RNA polymerase II-mediated general transcription but also function as specialized co-regulators in modulating distinct transcription programs critical for leukemogenesis.

RUNX1 activates genes, such as PU.1 which is important for the differentiation of hematopoiesis stem/progenitor cells[42], but it also represses gene expression. The same holds true for AE. In AE-expressing AML cells, RUNX1 and AE co-occupy similar locations in the genome with greater than 70% frequency[43,44].

The ETO portion of AE interacts with a nuclear receptor corepressor (N-CoR), Sin3a, and HDAC complex to repress transcription[45,46], but both p300 and N-CoR share similar genomic locations with RUNX1 and AE, at both AE upregulated and downregulated genes[44]. The co-existence of opposing transcription factors, and opposing co-activators and co-repressors at same location reveals the dynamic binding of these proteins and the need for fine tuning of AE target gene transcription. This notion is further supported by studies showing that the ratio of RUNX1 to AE may determine whether genes are upregulated or downregulated by AE[43]. Nonetheless, the regulatory mechanisms controlling the dynamic binding of AE at target loci remains incompletely understood. Our data that KD of TAF1 impairs AE recruitment at chromatin, provide a regulatory mechanism for AE binding at target loci.

While AE plays a pivotal role in AML development, it has no enzymatic domains that can be targeted for therapeutic purpose. We have pursued several strategies to target AE, for instance, targeting the p300 histone acetyltransferase (HAT) which enables AE to function as an AML-inducing oncogene, with salicylate and diflunisal, to inhibit the growth of AE-expressing leukemia cells[47]. Peptides that block the formation of the AE transcription factor complex (AETFC), which is also required for AE-driven leukemia, impede leukemogenesis[11]. In addition, targeting *ID1*, an AE target gene that participates in AKT activation[48], also has therapeutic potential for AE-expressing AML. While we now show that targeting TAF1 bromodomains to control AE-related leukemia is a potential therapeutic strategy, further investigation is required to determine the usefulness of TAF1 inhibitors in the treatment of AML.

## Methods

**Cell lines**. Kasumi-1 cells were grown in RPMI 1640 media supplemented with 20% fetal bovine serum (FBS), SKNO-1 cells were grown in RPMI 1640 media with 10% FBS and 10 ng/ml GM-CSF. K562 cells were grown in IMDM media with 10% FBS. OCI-AML3 were grown in alpha-MEM media plus 20% FBS. Kasumi-1 (catalog # CRL_2724) and K562 (catalog # CRL_3343) were purchased from ATCC. OCI-AML3 (catalog # ACC-582) and SKNO-1 (catalog # ACC-690) were purchased from DSMZ.

**Chemicals**. Bromodomain inhibitors Bay-364 (catalog no. SML1783), Bay-299 (catalog no. SML1756), and JQ-1 (catalog no. SML1524) were purchased from Sigma. (https://www.sigmaaldrich.com/catalog/product/sigma/sml1783?lang=en®ion=US).

**Lentivirus, retrovirus production, and concentration**. Lentiviruses were produced in 293T cells using lipofectamine 2000 as transfection reagent and psPAX2 and VSVG as packaging plasmids. Retroviruses were produced using Calcium Phosphate Transfection Kit from Sigma-Aldrich following the manufacturer's instruction. Viruses were collected 48 and 72 h after transfection and concentrated by lenti-X concentrator or retro-X concentrator (Clontech).

**Plasmid construction**. The pLKO.1 plasmid expressing two human TAF1 shRNAs (hTAF shRNA#1 and hTAF1 shRNA#2) and two mouse TAF1 shRNAs (mTAF1 shRNA#1 and mTAF1 shRNA#2) were purchased from Sigma. MigR1, MigR1-AML1-ETO (AE), and its amino acid mutation plasmids were described in Wang et al.[23]. TAF1 cDNA was purchased from Addgene; the full-length sequence was corrected and verified by DNA sequencing and reconstructed into the pCDH-MSCV-EF1 vector, purchased from SBI Biotech. The TAF1 bromodomain deletion (ΔBr) construct was cloned by deleting amino acids 1397–1510, using PCR-based mutagenesis.

**BrdU assay**. BrdU assay was performed using BD Pharmingen BrdU Flow Kit. Briefly, Kasumi-1 cells and CD34+ cells were transduced with scrambled shRNA and hTAF1 shRNAs. Four days after transduction, cells were incubated with 10 μM BrdU for 1 h. After fixation and permeabilization, cells were digested by DNase at 37 °C for 1 h. Following anti-BrdU staining, BrdU incorporation was analyzed by flow cytometry.

**Subcellular fractionation assay**. Subcellular fractionation assay of Kasumi-1 cells was performed using the Subcellular Protein Fractionation Kit for cultured cells (Thermo Scientific) according to the manufacturer's instruction. Briefly, Kasumi-1 cells were transduced with either scrambled or TAF1 shRNAs for 3 or 5 days. Equal number of cells from each treatment were collected and used for protein fractionation. Cellular proteins were fractionated to cytoplasmic, nuclear soluble, chromatin bound, and nuclear insoluble proteins. β-Actin and histone H3 were used for loading control of each fraction.

**Co-immunoprecipitation, western blot and antibodies**. Co-immunoprecipitation (co-IP) was performed in NETN buffer as described previously[49]. In brief, cell pellet was lysed in NETN buffer (50 mM Tris pH = 7.5; 150 mM NaCl; 1 mM EDTA; 1% NP40, phosphatase inhibitor, and protease inhibitor cocktail purchased from Roche) with sonication and then incubated at 4 °C for 1 h. Insolvable debris was removed by centrifugation. Following preclearing, an equal amount of cell lysate was subjected to the incubation with antibodies overnight at 4 °C. Magnetic protein A/G beads were added to precipitate protein–antibody complex. After four washes in NETN buffer, immunoprecipitated proteins were eluted with Laemmli protein sample buffer. Equal volume of co-IP samples were subjected to 4–12% premade polyacrylamide gels (Invitrogen). Antibodies used for western blots are listed in Supplementary Table 2. The unprocessed scans of the important blots are shown in the Supplementary Figs. 5–9.

**Flow cytometry**. To monitor the expression of cell surface markers c-Kit, Sca-1, Mac-1, and Gr-1, cells were stained by APC-conjugated c-Kit, PE-cy7-conjugated Sca1, PE-conjugated Mac-1, and percp-cy5.5-conjugated Gr-1 antibodies purchased from BD Biosciences. To monitor apoptosis, cells were stained with PE-conjugated annexin V and 7-AAD using PE Annexin V Apoptosis Detection Kit I from BD Biosciences. Stained cells were evaluated using FACS Canto-II and data were analyzed by FlowJo_V10 software.

**Chromatin immunoprecipitation and ChIP-sequencing**. Chromatin immunoprecipitation (ChIP) assays were performed using SimpleChIP Enzymatic Chromatin IP kit (Cell Signaling Technology) following the manufacturer instructions. In brief, cells were fixed and lyzed in ChIP buffer. After sonication, insoluble debris was removed by centrifugation. Ten percent of each supernatant was used as input. Remaining supernatant was diluted in ChIP buffer and incubated with antibody overnight at 4 °C. Magnetic protein A/G beads precoated with sperm DNA were added for 1 h before extensive washes. Immunoprecipitated chromatin fragments were digested with proteinase K and the crosslink between DNA and proteins was reversed at 65 °C for 2 h. DNA was isolated by either spin columns or phenol/chloroform extraction and quantitated by reverse transcriptase-PCR or subjected for ChIP-sequencing library preparation.

ChIP-sequencing was performed at Oncogenomic Core Facility at the Sylvester Comprehensive Cancer Center. IP samples and input were sequenced using single-end reads with an Illumina NextSeq 500. Reads were trimmed for adapters using Cutadapt [v1.15] –nextseq-trim = 20 –m 18. Fastq files were aligned to human GRCh38.p3 using BWA [v0.7.13] with parameters aln -q 5 -l 32 -k 2. AE peaks were called using macs2 v (2.1.1.20160309) that pass an Irreproducible Discovery Rate (IDR) < 0.05 between two biological replicates. TAF1 peaks were determined by overlapping peaks with IDR < 0.05 for at least two out of three biological replicates. Shift and extension sizes were determined using phatompeakqualtools (v1.1). ChIPseeker v1.14.0 and R 3.4.1 were used for peak annotation. Peak overlaps were determined by merging a master peak list from a given comparison and intersecting individual peak files to the master files by at least 1 bp. Peaks were considered to be overlapping if samples overlapping the same peak in the master peak list. Motif analysis was performed using MEME-ChIP v4.12.0 and then JASPAR 2018 motif database. The accession number is GSE100446.

To evaluate the effect of TAF1 knockdown on AE binding, Kasumi cells, transduced with scrambled shRNA or TAF1 shRNA, were sonicated with Diagenode Pico for 14× 30 s on/off cylces. Twenty micrograms of sonicated chromatin samples and an antibody against AML1-ETO (Invitrogen, PA5-40076) were used for ChIP (Magna ChIP HiSens Chromatin Immunoprecipitation kit, Millipore 17-10460) with two additional high-salt buffer washes. ChIP DNA was purified and DNA sequencing library was constructed using NEBNext Ultra II DNA library Prep Kit (NEB E7645). The DNA sequencing was performed on a Nextseq 500 from Illumina. The DNA sequences were aligned to the human genome (hg19) using bowtie2 (2.2.6) with the default parameters. DeepTools (3.1.3) was used to compare the AML1-ETO binding at these shared binding sites after the normalization of the ChIP-sequencing data. AML1-ETO binding peaks were called with MACS2 (2.1.1.20160309) and 6695 common peaks were derived from two independent assays. In all, 6126 TAF1 peaks were lifted from hg38 to hg19 and 1061 sites among them are shared with AML1-ETO peaks.

**ATAC-sequencing**. ATAC-sequencing was performed as described previously[50]. Briefly, Kasumi-1 cells transduced with scrambled shRNA or TAF1 shRNAs were collected and lyzed in lysis buffer (10 mM Tris-HCl, pH 7.4, 10 mM NaCl, 3 mM MgCl$_2$, 0.1% IGEPAL). Transposition reaction was performed using Nextera Tn5 Transposase kit (Illumina). After PCR purification, DNA was amplified using custom Nextera PCR primers. ATAC-seq chromatin accessible regions were determined using ENCODE pipeline standards (https://github.com/ENCODE-

DCC/atac-seq-pipeline; git commit 2b693abd4550943be1e8d9a686a1050c1a-cab92c). Briefly, sequencing indices were trimmed from merged fastq files using cutadapt (1.9.1) and then aligned to hg38 using bowtie2 (2.2.6). After de-duplication, reads were tn5 shifted and replicate peaks were called and fold enrichment signal files were generated using macs2 (2.1.0). Final peaks were determined using IDR (2.0.4) of true replicates. Heatmaps and profile plots were generated using deeptools (3.1.1). TSS and enhancers were annotated using ChIPSeeker and R v3.4.1 and filtered for regions annotated to RNA-seq differentially expressed genes. TSS was defined as ±3 kb surrounding Ensembl annotated transcription start sites and enhancers were defined as the regions with both H3K4me and H3K27Ac peaks excluding TSS regions. All samples passed ENCODE3 accessibility quality standards as measured by ATAQC—indicating enrichment at TSS regions and the ability to distinguish multi-nucleosomes from fragment length.

**Mass spectrometry**. The co-immunoprecipitation were performed as described above using anti-TAF1 or anti-ETO antibodies in Kasumi-1 cells. The immuno-precipitated proteins were separated by 4–12% Bis-Tris NuPAGE gel and the gel was stained using coomassie blue. Excised gel pieces were sent to Taplin Mass Spectrometry Facility at Harvard Medical School for further sample preparation and analysis. Unique indicates the number of unique peptide matches to the protein; total indicates the total number of peptides matched to the protein. AVG indicates the average Xcorr value for all the peptides matched to the protein. Xcorr is a score used by Sequest (the search algorithm) to judge the quality of the spectral match to a particular peptide sequence.

**RNA isolation, quantitative PCR, and RNA-sequencing**. RNA was extracted using RNeasy mini kit (Qiagen) and cDNA was generated by QuantiTect Rev. Transcription Kit from Qiagen following the manufacturer's instructions. The thermal cycle conditions to amplify cDNA were 48 °C for 15 min; 95 °C for 10 min, followed by 40 cycles of 95 °C for 15 s; 60 °C for 1 min, and 18S rRNA was used as an internal control. The sequences of primers or the catalog numbers of TaqMan primers are summarized in Supplementary Table 3.

In total, six samples were collected from TAF1 knockdown Kasumi-1 cells, comprised of two distinct shRNAs and two independent experiments, and five samples were collected from cells infected with scrambled shRNA; four samples were collected from either AE knockdown Kasumi-1 cells or cells infected with scrambled shRNA, comprised of two distinct shRNAs; four samples were collected from Kasumi-1 cells either treated with vehicle (DMSO) or 10 μM Bay-364 for 3 days. RNA was extracted using RNeasy plus micro kit (Qiagen). Library preparation and RNA-sequencing were completed at Oncogenomic Core Facility at the Sylvester Comprehensive Cancer Center. Samples were sequenced using paired ends with an Illumina NextSeq 500 and subsequent sequencing reads were trimmed and filtered using Cutadapt (see ChIP-seq data analysis). Fastq files were aligned to Ensembl 87: GRCh38.p7 human transcriptome using STAR aligner [v2.5.3a] and RSEM [v1.3.0] to obtain expected gene counts. GC nomination between sequencing runs was performed using EDAseq v.2.12.0 within lane GC normalization. Differential expression was determined between TAF1 shRNA or AE shRNA and scrambled shRNA using DESeq2 [v1.30.0] and R [v3.4.1] with a Benjamini–Hochberg FDR cutoff of 0.05. Heatmaps were generated using euclidean distances between sample blind, variance stabilized transformed counts from DESeq2. The accession number is GSE100446.

**CB CD34+ cell purification**. Human CB was purchased from the New York Blood Center. The purification of CD34+ cells was described previously[23]. Briefly, white blood cells were separated from other types of blood cells by centrifugation. After the wash, CD34+ cells were magnetically labeled with CD34+ MicroBeads and then were separated using MACS column. Human CD34+ MicroBead kit was purchased from Miltenyi Biotec Company.

**Colony forming assays and serial replating assays**. Purified CD34+ cells were transduced with scrambled shRNA or TAF1 directed shRNA followed by puromycin selection for 48 h and then transduced with GFP+ MigR1 or MigR1-AE viruses for 2 days. GFP+ CD34+ cells were sorted using FACS Aria Ilu (BD Biosciences). CD34+ cells expressing either MigR1 or MigR1-AE were resuspended in Methocult H4230 (human cells) and replated into six-well plates, with a density of 3000 cells per well for colony assay. Bone marrow cells isolated from Mx1-Cre mice or AE knock-in mice were transfected with scrambled shRNA or TAF1 directed shRNAs followed by puromycin selection for 48 hours. 3000 cells from each transfected group were resuspended and plated in Methocult GF M3434, seven days after the initial plating, colonies were counted, then all the cells were collected for replating weekly at a density of 3000 cells per well for 4 weeks.

**CAFC assays**. CAFC assays were performed as described previously[23] To perform CAFC asays, bone marrow cells were seeded onto MS5 cells in MEM alpha medium containing 12.5% FBS, 12.5% horse serum, and 1 μM hydrocortisone. Medium was semi-replenished every week and number of cobblestone area was counted after 5 weeks.

**Liquid culture differentiation assays**. To monitor myeloid differentiation, sorted GFP+ CD34+ cells were grown in myeloid differentiation medium X-VIVO supplemented with 20% BIT 9500 (Stem Cell Technologies), SCF 100 ng/ml, FLT3-L 10 ng/ml, IL-3 20 ng/ml, G-CSF 20 ng/ml, GM-CSF 20 ng/ml, and IL-6 20 ng/ml (PeproTech) for 4 days, the expression of the myeloid differentiation marker Mac-1 was measured by flow cytometry.

**Bone marrow cell isolation**. Poly(I:C) (10 mg/kg every other day × 3 was administered to Mx1-Cre and AE knock-in mice to induce AE expression. Ten days after induction, tail-vein blood was collected to assess AE expression. Four days before isolating bone marrow stem cells, 5-FU (150 mg/kg) was injected into the mice to deplete proliferating blood cells. Two weeks after poly(I:C) application, mice were sacrificed and bone marrow cells were isolated from both femurs and tibias. During the preparation of bone marrow cells, ACK (ammonium-chloride-potassium) lysing buffer was used to lyse red blood cells. Isolated bone marrow cells were plated in IMDM medium supplemented with 10% fetal bovine serum and cytokines (SCF 100 ng/ml, IL-6 20 ng/ml, and IL-3 10 ng/ml).

**Leukemia mouse models**. Fetal liver (FL) cells were isolated from E14.5 embryos of C57BL/6 mice and infected with GFP-tagged MigR1-AE9a retroviruses. GFP-positive FL cells were sorted by flow cytometry. Six to eight weeks female C57BL/6J recipient mice were purchased from The Jackson Laboratory and lethally irradiated with 950 cGy. After irradiation, recipient mice were transplanted with GFP-positive FL cells through tail-vein injection. After transplantation, primary leukemia cells were collected from the spleen of mice at the end point and injected into a second batch of recipient mice for secondary transplantation. Secondary spleen leukemia cells were collected from mice at the end point after secondary transplantation. Primary spleen cells were also grown in RPMI 1640 with 20% fetal bovine serum for weeks to develop AE9a+ cell line. Luciferase gene was integrated into AE9a+ cell line to create AE9a+ luciferase+ cell line. AE9a+ luciferase+ cell line or secondary spleen leukemia cells were transduced with scrambled shRNA or mTAF1 shRNA#1 or mTAF1 shRNA#2. After confirming the knockdown of TAF1, the secondary spleen leukemia cells or AE9a+ luciferase+ cell line were injected into C57BL/6 recipient mice which have received sublethal irradiation (450 cGy). Three weeks after transplantation, the percentage of GFP+ AE9a+ luciferase+ cell line in the peripheral blood was monitored by flow cytometry every week. The distribution of luciferase-positive AE9a+ cells in the mice was monitored using IVIS imaging system weekly. No statistical method was used to predetermine the number of mice in each group. All mice were assigned into each group randomly and the investigators were not blinded during the experiments and outcome assessment.

All animal studies were conducted in accordance with NIH guidelines for the care and use of laboratory animals and were approved by the IACUC of the University of Miami.

**Statistical Analysis**. Significance were calculated using unpaired two-tailed Student's t-test. Survival curves were analyzed using Kaplan–Meier method in GraphPad Prism 6.0 software.

**Reporting summary**. Further information on research design is available in the Nature Research Reporting Summary linked to this article.

## Data availability
RNA-seq, ChIP-seq, and ATAC-seq data have been deposited in the NCBI Gene Expression Omnibus (GEO) database, accession code: GSE100446. All other remaining data are available within the article or supplementary files or are available upon request from the corresponding author S.N. (snimer@med.miami.edu).

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

## Acknowledgements

We thank the members of Dr. Nimer's lab for their technical support and helpful suggestions and Delphine Prou for her assistance in animal study. We also thank Bernard Jay Wasserlauf for his technical support on IVIS, the Flow Cytometry Shared Resources for sorting and flow cytometry analysis, Oncogenomic Shared Resources and Bioinformatics and Biostatistics Shared Resources at Sylvester Comprehensive Cancer Center for RNA-sequencing and ChIP-sequencing library preparation and sequencing. We also thank Dr. Stefan Kubicek at CeMM Research Center for Molecular Medicine of the Austrian Academy of Sciences in Australia, Dr. Hartmut Geiger at Ulm University in Germany, and Dr. Bob Roeder at Rockefeller University for generously providing TAF1 antibodies. Alejandro Roisman is a Fellow of The Leukemia & Lymphoma Society. This project was supported by a grant from National Cancer Institute (R01CA166835) to S.N. and a grant from American Cancer Society (IRG-17-183-16) to Y.X.

## Author contributions

Y.X. and S.N. conceived the project, designed the experiments, and wrote the manuscript. Y.X. conducted most of the experiments. N.M. performed the IVIS experiments. C.M. generated plasmids, C.J.M. performed co-immunoprecipitation assays, F.L. performed subcellular fractionation assays, D.K., J.S., G.M.M., and F.B. performed bioinformatics analysis for RNA-seq and ChIP-seq. F.L. and J.Y. performed library preparation for RNA-seq and ChIP-seq. S.D. performed real-time PCR. A.R. helped with CRISPR. S.G. isolated CD34+ cells. L.W. provided AE9a+, luciferase+ cells. X.S. performed p300 ChIP-seq. M.F. performed patient data analysis. R.S. provided advice and edited the manuscript.

## Competing interests

The authors declare no competing interests.
