## [Peer Review File · Nature Communications]

Reviewers' comments:

Reviewer #1 (Remarks to the Author):

This manuscript by Xu et al describes the function of TAF1 in AML-ETO induced myeloid leukemias. This is an interesting and important question, and the clearly written manuscript presents strong experimental evidence of the requirement of TAF1 for AML1-ETO induced AML. The experimental models are well developed, data are of high quality, and results are thoughtfully interpreted.

While further questions can be asked to better understand the biochemical basis of the AE-TAF1 binding, its pharmacologic blockade, and the functional effects on gene expression and chromatin structure, I believe that these questions are beyond the scope of the current, already extensive manuscript. However, I would suggest expanding the discussion of the possible mechanisms and functions by which the AE-TAF1 interaction regulates gene expression, with attention to the following questions:

1) TAF1 is best recognized as a component of TFIID with functions at promoter regions, but as the authors note, a fraction of TAF1 is associated with chromatin outside of promoter regions, and AE itself is mostly localized away from gene promoters. Does this mean that TAF1-AE interaction is restricted to promoter-bound complexes?

2) If this interaction also occurs outside of promoters, then what hypothetical functions could TAF1, either in complex with other TFIID factors, or another complex entirely, could it exert outside of promoters? Is it likely to involve ectopic transcription and consequent chromatin disruption?

3) Recently, Vakoc and colleagues reported a dependency of a variety of AML cell lines on TAF12 (Cancer Cell 2018). Could these two sets of findings be related, and could TAF1 dependency occur in other AML subtypes in addition to AML1-ETO?

4) TAF1 depletion is associated with partial eviction of AML1-ETO from chromatin to nucleoplasm (Fig. 7a). How can this be reconciled with the fact that AML1-ETO can bind to DNA directly?

5) The authors may wish to discuss possible technical reasons why TAF1 shRNA depletion is nicely observed in total cell extracts and chromatin fractions, but not in cytoplasmic or nucleoplasmic fractions (Fig. 7a).

Reviewer #3 (Remarks to the Author):

In Xu et al., the authors propose that TAF1 binds to acetylated K43 on the AML1-ETO fusion transcription factor, and that this recruitment is required for AML1-ETO mediated gene expression, recruitment to chromatin, and ultimately growth of AML1-ETO leukemia. The authors show that other cell types are less dependent on TAF1 suggesting a therapeutic index and that TAF1 KD in AML1-ETO fusion leukemic cells blocks tumor growth and increases overall survival.

Overall this is a nice manuscript with strong phenotypic data. These data provide additional support that targeting of the transcriptional apparatus can be selective especially in tumors that harbor underlying deregulation of transcription.

In its current form, the mechanistic conclusions of the manuscript need to be bolstered. For instance, despite the clear logic linking P300 acetylation of AML1-ETO to TAF1 recruitment, and the well studied role of TAF1 in promoting transcription initiation, the authors suggest a unique function for TAF1 in AML1-ETO where it is able to both activate and repress genes. No evidence is

provided to suggest what exactly this function is other than a requirement for the K43 acetyl to TAF1 bromodomain interaction. Given TAF1's various non-PIC bridging functions (its two kinase domains, acetyltransferase activity, and ubiquitin activation/conjugation), the paper's claim that TAF1 is targetable in AE-AML would be significantly strengthened by demonstrating that one of these catalytic functions is actually critical to their proposed mechanism. Mutation studies—as opposed to large deletion studies—targeting the catalytic domains would be helpful. It is also possible that the large numbers of observed up regulated and down regulated genes upon TAF1 or AML1-ETO perturbation result from a failure to properly interpret gene expression analysis that assume no global changes in gene expression. TAF1, as a general transcription factor is almost certain to have a global role in gene expression and this is masked by the author's analysis.

Finally, the ChIP-seq analysis is mostly limited to overlap analysis and could be further explored to support the mechanisms being proposed. For instance ChIP-seq of AML1-ETO after acute TAF1 depletion would help strengthen the claim that AML1-ETO chromatin recruitment is reduced upon TAF1 KD. The current analysis in Figure 7A needs to show that other chromatin associated proteins are not affected by TAF1 KD.

Reviewer #1 (Remarks to the Author):

This manuscript by Xu et al describes the function of TAF1 in AML-ETO induced myeloid leukemias. This is an interesting and important question, and the clearly written manuscript presents strong experimental evidence of the requirement of TAF1 for AML1-ETO induced AML. The experimental models are well developed, data are of high quality, and results are thoughtfully interpreted.

While further questions can be asked to better understand the biochemical basis of the AE-TAF1 binding, its pharmacologic blockade, and the functional effects on gene expression and chromatin structure, I believe that these questions are beyond the scope of the current, already extensive manuscript. However, I would suggest expanding the discussion of the possible mechanisms and functions by which the AE-TAF1 interaction regulates gene expression, with attention to the following questions:

Question 1.1 TAF1 is best recognized as a component of TFIID with functions at promoter regions, but as the authors note, a fraction of TAF1 is associated with chromatin outside of promoter regions, and AE itself is mostly localized away from gene promoters. **Does this mean that TAF1-AE interaction is restricted to promoter-bound complexes?**

Answer 1.1 We thank the reviewer for asking clarification regarding whether the TAF1-AE interaction is restricted to promoter bound complexes. Based on our ChIP-seq data, a majority of TAF1 peaks are present at promoter regions, while AE peaks are distributed at promoter regions, distal intergenic regions and within introns. We estimated the p values using a Monte Carlo simulation of shuffled peaks within either the TSS background or the nonTSS genomic background and found that TAF1 and AE shared binding sites are located at both TSS ($p < 1.0e-5$) and non-TSS regions ($p < 1.0e-3$), implying that the interaction of TAF1 and AE is not restricted to promoter regions. In addition, we performed KEGG analysis for those unique AE peaks which

are not overlapping with TAF1 at TSS and non-TSS. AE unique peaks at TSS have no significant KEGG enrichment for

annotated genes and AE unique peaks at non-TSS are not directly involved in “pathways in cancer” or “acute myeloid leukemia” (Supplementary Figure 4f). We have added these data and explanations to the results section and the second paragraph of the discussion section.

Question 1.2 If this interaction also occurs outside of promoters, then what hypothetical functions could TAF1, either in complex with other TFIIID factors, or another complex entirely, could it exert outside of promoters? Is it likely to involve ectopic transcription and consequent chromatin disruption?

Answer 1.2 We thank the reviewer for raising up these questions. We discuss the role of TAF1 outside of promoters. To better define the role of TAF1 in regulating the function of AE (or possibly other transcriptional regulators) at non-promoter regions, we performed additional experiments.

Supplementary Figure 2e

unique	total	reference	gene symbol
12	30	Q06455_MTG8_HUMAN	RUNX1T1
13	19	Q01196 RUNX1_HUMAN	RUNX1
8	9	Q13951 PEBB_HUMAN	CBFB
2	2	P21675 TAF1_HUMAN	TAF1
2	2	Q92804 RBP56_HUMAN	TAF15

The data are included in the results and discussion sections.

For instance, we performed coIPs in Kasumi-1 cells using an anti-ETO antibody and then detected those proteins associated with AE by western blot and mass

spectrometry. In addition to TAF1, we can only detect TAF15, but not other components of TFIID (Supplementary Figure 2d and 2e). It is well known that TAF7 forms a subcomplex with TAF1 in the TFIID complex, and we found TAF7 binding to the AE peptide in our 2011 paper. However, TAF7 was not identified in the recently conducted IP experiment. Studies using human embryonic stem cells have shown that the combination of TAFs in the PIC is both cell context and promoter dependent ¹. Thus, while we can not exclude that limitations in the affinity and specificity of the antibodies used or the co-immunoprecipitation protocol hampered our ability to identify other TAFs in the AE/TAF1 complex, it appears that few TAFs are engaged in the AE/TAF1 complex. We also analyzed the AE/TAF1 overlapping peaks and the AE unique peaks at non-TSS regions and found that the AE/TAF1 overlapping peaks are adjacent to genes involved in “pathways in cancer” and “acute myeloid leukemia” (Supplementary Figure 4e) while AE unique peaks are not directly related to these pathway (Supplementary Figure 4f). ENCODE and ChEA Consensus TFs analysis reveals that MYC is only found at those genes adjacent to AE/TAF1 shared sites at TSS, not at those genes adjacent to AE unique sites at TSS. The functional implications of this difference requires further investigation. Our RNA-seq data indicates that TAF1 controls the

transcription of a subset of AE upregulated and downregulated genes. However, the expression of RNA polymerase II dependent housekeeping genes such as *ACTB*, *GAPDH* and cell cycle regulatory genes such as *CCND2*, *CCND3* and *CDKN2D* was not affected by TAF1 knockdown (RNA polymerase I dependent ribosomal transcript 18S served as the internal control)

(Supplementary Figure 4c). This suggests that the residual TAF1 in our system was sufficient for its role as a general transcription factor. In addition, we performed ATAC-seq in Kasumi-1 cells

with normal or reduced levels of TAF1 and found that TAF1 KD does not alter chromatin

accessibility globally (Supplementary Figure 4h) which would suggest that TAF1 promotes the more local functions of AE. Further, we analyzed eRNA expression at AE/TAF1 co-bound sites and found that TAF1 KD does not significantly affect eRNA expression globally (see picture on the left). In summary, it is likely TAF1 serves as a specific co-activator or a co-repressor for AE mediated transcription. We have included our answers to these questions in

the highlighted portion of the first, second, third and fourth paragraphs in the discussion (and in Supplementary Figure 2d, 2e, 4c, 4e, 4f and 4h).

3) Recently, Vakoc and colleagues reported a dependency of a variety of AML cell lines on TAF12 (Cancer Cell 2018). Could these two sets of findings be related, and could TAF1 dependency occur in other AML subtypes in addition to AML1-ETO?

Answer 1.3 The reviewer has brought up an important point as more and more TAFs have been recently discovered to be involved in leukemia. For instance, TAF1 has been found in both CBF β -MYH11 and RUNX1 containing complexes². The rearrangement of TAF15 with ZNF384 has been found in AML and ALL, and the fusion protein is thought to play a role in progenitor cell differentiation³. CRISPR screening indicates that individual TAFs within TFIID are required for the growth of different leukemic cells. Together with Dr. Vakoc's discovery, all of these studies suggest that TAFs not only participate in RNA polymerase II mediated transcription but also

function as specialized co-regulators in modulating transcription. We point out that CRISPR screening has found TAF1 is essential for the growth of MV4-11 cells, but not HL-60, MOLM-13 or OCI-AML3⁴. Although the definite roles of TAF1 in MLL-AF4 or CBF β -MYH11 expressing AML remains unclear, these studies reveal the importance of TAFs in modulating cell behavior in subsets of acute leukemia. Our detailed answer to this query is contained in the sixth paragraph of the discussion.

Question 1.4 TAF1 depletion is associated with partial eviction of AML1-ETO from chromatin to nucleoplasm (Fig. 7a). How can this be reconciled with the fact that AML1-ETO can bind to DNA directly?

Answer 1.4 We thank the reviewer for asking us to address this question, which we now address in the fourth paragraph of our discussion. Our ChIP-seq data clearly indicate sites in the genome where AE binds without TAF1 and others where they co-localize. The basis for this difference is being explored. Although AE can bind to DNA directly, the recruitment of AE on the chromatin is regulatable. For instance, CBF β enhances the binding of RUNX1 to its DNA consensus binding sequence⁵. Given that TAF1 HAT domain acetylates histone H3 and H4 and histone acetylation

may alter local chromatin structure, we investigated whether TAF1 HAT domain assists in the recruitment of AE to chromatin. We performed subcellular fractionation of Kasumi-1 cells sequentially infected with viruses expressing TAF1 shRNA and HA-tag TAF1 wildtype or HA-tag TAF1 HAT domain deletion. The HAT domain deletion restores the recruitment of AE, overcoming the effects of TAF1 KD

similar to TAF1 wildtype proteins (see Fractionation figure). Thus the TAF1 HAT domain does not demonstrably influence AE recruitment on chromatin. TAF1 also has two DNA binding modules, two kinase domains and one ubiquitin ligase domain; the functionality of these domains is not universally accepted, making it difficult to assay the function of deletion mutants. Besides their functional capacity, these domains may promote the binding of AE to chromatin. As a demonstrable scaffold protein, TAF1 domains may help provide a platform for the assembly of other proteins which could assist in the recruitment of AE to chromatin.

Question 1.5 The authors may wish to discuss possible technical reasons why TAF1 shRNA depletion is nicely observed in total cell extracts and chromatin fractions, but not in cytoplasmic or nucleoplasmic fractions (Fig. 7a).

Answer 1.5 We would thank the reviewer for asking that we address this issue. In previous Figure

7a, the TAF1 western blot for cytoplasm and nucleoplasm fractions was overexposed. In the new Figure 7a, we added the shorter exposure of TAF1. Although we could not detect a chromatin fraction of TAF1 in the shorter exposure, we can see

the knockdown of TAF1 protein levels in both the cytoplasm and nucleoplasm of a shorter exposure. Since the majority of TAF1 is located in the cytoplasm and nucleoplasm, the influence of TAF1 KD will be easiest to visualize where there is the least amount of TAF1 (on chromatin).

Reviewer #3 (Remarks to the Author):

In Xu et al., the authors propose that TAF1 binds to acetylated K43 on the AML1-ETO fusion transcription factor, and that this recruitment is required for AML1-ETO mediated gene expression, recruitment to chromatin, and ultimately growth of AML1-ETO leukemia. The authors show that other cell types are less dependent on TAF1 suggesting a therapeutic index and that TAF1 KD in AML1-ETO fusion leukemic cells blocks tumor growth and increases overall survival. Overall, this is a nice manuscript with strong phenotypic data. These data provide additional support that targeting of the transcriptional apparatus can be selective especially in tumors that harbor underlying deregulation of transcription.

Question 3.1 In its current form, the mechanistic conclusions of the manuscript need to be bolstered. For instance, despite the clear logic linking P300 acetylation of AML1-ETO to TAF1 recruitment, and the well studied role of TAF1 in promoting transcription initiation, the authors suggest a unique function for TAF1 in AML1-ETO where it is able to both activate and repress genes. No evidence is provided to suggest what exactly this function is other than a requirement for the K43 acetyl to TAF1 bromodomain interaction. Given TAF1's various non-PIC bridging functions (its two kinase domains, acetyltransferase activity, and ubiquitin activation/conjugation), the paper's claim that TAF1 is targetable in AE-AML would be significantly strengthened by demonstrating that one of these catalytic functions is actually critical to their proposed mechanism. Mutation studies —as opposed to large deletion studies— targeting the catalytic domains would be helpful.

Answer 3.1 The reviewer brought up a very interesting question and to answer this question, we used several approaches:

1) **ATAC-seq:** To investigate whether TAF1 KD alters chromatin accessibility globally, we

performed ATAC-seq using Kasumi-1 cells infected with scrambled shRNA or TAF1 shRNAs. TAF1 KD does not change the chromatin accessibility globally (see Supplementary Figure 4h) which would suggest that TAF1 promotes the more local functions of AE.

2) **Subcellular fractionation assays:** We infected Kasumi-1 cells first with viruses expressing scrambled shRNA or TAF1 shRNA and then with lenti-viruses of PCDH empty vector or HA-tagged TAF1 wildtype or TAF1 δ HAT (deletion of amino acid from 517 to 976). 96 hours after transduction, cells were collected for subcellular fractionation assay using the Subcellular Protein Fractionation Kit from Thermo Fisher Scientific, in accordance with the manufacturer's

instructions. As shown in the fractionation figure, we could observe the reduction of AE on chromatin in Kasumi-1 cells transduced with TAF1 shRNA viruses. Overexpression of HA-TAF1 wild type partially reverses the reduction of AE binding triggered by TAF1 KD, and overexpression of the TAF1 HAT domain deletion protein also reverses the decrease of AE binding on chromatin, defining that the TAF1 HAT

domain is not required for the chromatin association of AE. In the future, we will continue to investigate whether other TAF1 enzymatic domains or putative DNA binding modules are responsible for the AE recruitment or that TAF1 provides a platform for other proteins which could assist in the recruitment of AE onto chromatin.

3) **EMSA assays:** We performed EMSAs to identify whether specific domains in TAF1 were critical for modulating AE binding to DNA. However, AE/TAF1 complex bound DNA is too big to be detected by this assay, even using a 4.5% polyacrylamide gel.

Question 3.2 It is also possible that the large numbers of observed up regulated and down regulated genes upon TAF1 or AML1-ETO perturbation result from a failure to properly interpret gene expression analysis that assume no global changes in gene expression. TAF1, as a general transcription factor is almost certain to have a global role in gene expression and this is masked by the author's analysis.

Answer 3.2 To determine whether TAF1 knockdown globally suppresses RNA polymerase II

dependent transcription in our cells, we compared a panel of RNA Polymerase II dependent housekeeping genes such as *ACTB* and *GAPDH* and cell cycle regulatory genes such as *CCND2*, *CCND3*, *CDKN2D* in TAF1 knockdown conditions using the RNA Polymerase I dependent ribosomal transcript 18S as internal control. We found that TAF1 KD does not

broadly influence RNA Polymerase II dependent transcripts, suggesting that residual TAF1 in our system was sufficient for its role as a general transcriptional regulator.

Question 3.3 Finally, the ChIP-seq analysis is mostly limited to overlap analysis and could be further explored to support the mechanisms being proposed. For instance ChIP-seq of AML1-ETO after acute TAF1 depletion would help strengthen the claim that AML1-ETO chromatin

recruitment is reduced upon TAF1 KD. The current analysis in Figure 7A needs to show that other chromatin associated proteins are not affected by TAF1 KD.

Answer 3.3. We have addressed these two key points and have performed AE CHIP-seq in

Kasumi-1 cells transduced with scrambled shRNA or TAF1 shRNA. We compared the intensity of the AE peaks that overlap with TAF1 peaks using DeepTools (3.1.3). (Supplementary Figure 4g) and found that the AE binding signal at AE/TAF1 co-

bound sites is significantly greater in Kasumi-1 cells with normal levels of TAF1 than in cells with reduced levels of TAF1 ($p < 3.3e-17$). Thus, both the subcellular fractionation assay (Figure 7a) and the ChIP-seq data show that TAF1 KD significantly reduces the recruitment of AE to chromatin.

To address the second concern, we have added the subcellular fractionation of PU.1 to Figure

7a. The data indicate that the association of PU.1 on chromatin is not affected by TAF1 KD, implying that the impact of AE binding at chromatin by TAF1 KD is specific.

REFERENCES

1. Maston, G.A. *et al.* Non-canonical TAF complexes regulate active promoters in human embryonic stem cells. *Elife* **1**, e00068 (2012).
2. Mandoli, A. *et al.* CFBF-MYH11/RUNX1 together with a compendium of hematopoietic regulators, chromatin modifiers and basal transcription factors occupies self-renewal genes in inv(16) acute myeloid leukemia. *Leukemia* **28**, 770-778 (2014).
3. Kim, J., Kim, H.S., Shin, S., Lee, S.T. & Choi, J.R. t(12;17)(p13;q12)/TAF15-ZNF384 Rearrangement in Acute Lymphoblastic Leukemia. *Ann Lab Med* **36**, 396-398 (2016).
4. Tzelepis, K. *et al.* A CRISPR Dropout Screen Identifies Genetic Vulnerabilities and Therapeutic Targets in Acute Myeloid Leukemia. *Cell Rep* **17**, 1193-1205 (2016).
5. Roudaia, L. *et al.* CBFbeta is critical for AML1-ETO and TEL-AML1 activity. *Blood* **113**, 3070-3079 (2009).

REVIEWERS' COMMENTS:

Reviewer #1 (Remarks to the Author):

The revised manuscript addresses all of my questions.

Reviewer #3 (Remarks to the Author):

The authors have largely addressed my concerns within the scope of what I believe to be reasonable for this manuscript.

The new figure S4G is difficult to interpret but does appear to support the author's point that TAF1 mediates AML1-ETO chromatin recruitment. A simpler presentation of this paired data would be to plot a histogram of the ratios, which I think will show a distribution well-removed from 1.0. Both replicates could be shown on one plot.

Overall, this is an elegantly conducted study with strong phenotypic data that supports targeting of TAF1 as a therapeutic strategy in AML1-ETO leukemias. I still think it is unclear how TAF1 actually functions at genes in the context of AML1-ETO. The author's data still mostly support a model where TAF1 plays an activating role in transcription. I would encourage the authors to follow up this work with further mechanistic dissection of TAF1 functional domains to better understand how TAF1/AML-ETO regulates transcription.

REVIEWERS' COMMENTS:

Reviewer #1 (Remarks to the Author):

The revised manuscript addresses all of my questions.

Reviewer #3 (Remarks to the Author):

The authors have largely addressed my concerns within the scope of what I believe to be reasonable for this manuscript.

The new figure S4G is difficult to interpret but does appear to support the author's point that TAF1 mediates AML1-ETO chromatin recruitment. A simpler presentation of this paired data would be to plot a histogram of the ratios, which I think will show a distribution well-removed from 1.0. Both replicates could be shown on one plot.

We thank the reviewer for suggesting we depict our data in a simpler format. As suggested, we

have generated a histogram plot depicting the ratio of AE binding to chromatin, in TAF1KD cells vs cells with normal TAF1 levels at the AE binding sites which are also occupied by TAF1. As shown in the new Supplementary Figure 4g, the AE binding ratio is smaller than 1, indicating that TAF1 KD reduces AE recruitment to chromatin.

Overall, this is an elegantly conducted study with strong phenotypic data that supports targeting of TAF1 as a therapeutic strategy in AML1-ETO leukemias. I still think it is unclear how TAF1 actually functions at genes in the context of AML1-ETO. The author's data still mostly support a model where TAF1 plays an activating role in transcription. I would encourage the authors to follow up this work with further mechanistic dissection of TAF1 functional domains to better understand how TAF1/AML-ETO regulates transcription.

We thank the reviewer for calling our study elegant, with strong phenotypic data. We can assure the reviewer that we will create additional TAF1 enzymatic domain deletion constructs and use them to identify which TAF1 enzymatic domain is critical for bringing AE to chromatin and for other functions of AE.